# VARIATIONAL QUANTUM ALGORITHMS ARE LIPSCHITZ SMOOTH

## ABSTRACT

The successful gradient-based training of Variational Quantum Algorithms (VQAs) hinges on the $L$-smoothness of their optimization landscapes—a property that bounds curvature and ensures stable convergence. While $L$-smoothness is a common assumption for analyzing VQA optimizers, there has been a need for a more direct proof for general circuits, a tighter bound for practical guidance, and principled methods that connect landscape geometry to circuit design. We address these gaps with Four core contributions. First, we provide an intuitive proof of L-smoothness and derive a new bound on the smoothness constant, $L \leq 4\|M\|_2 \sum_{k=1}^{P} \|G_k\|_2^2$, that is never looser and often strictly tighter than previously known. Second, we show that this bound reliably predicts the scaling behavior of curvature in deep circuits and identify a saturation effect that serves as a direct geometric signature of inefficient overparameterization. Third, we leverage this predictable scaling to introduce an efficient heuristic for setting near-optimal learning rates. Fourth we demonstrate that our heuristic remains robust in noisy environments enabling Adam and SGD to achieve convergence rates competitive with the Quantum Natural Gradient optimizer.

## 1 INTRODUCTION

Variational Quantum Algorithms (VQAs) are a cornerstone of near-term quantum computing Peruzzo et al. (2014), but a complete theoretical understanding of their optimization landscapes is still developing McClean et al. (2018). The performance of gradient-based optimization algorithms used to train these models fundamentally relies on the geometry of the objective function's landscape Schuld et al. (2019). A key property for guaranteeing the convergence of these optimizers is *L-smoothness*. This property has direct practical consequences: by placing an upper bound on the landscape's maximum curvature, it ensures that the gradient cannot change arbitrarily fast, preventing the unstable training dynamics that can arise from an optimizer overshooting sharp valleys. Crucially, a bound on the smoothness constant, $L$, provides a theoretical basis for learning rate selection moving this critical hyperparameter choice from a heuristic art to a more grounded science Nesterov (2013). Although this property has been discussed and global smoothness constants have been derived in prior work, the proofs often rely on bounding techniques that are less tight or less directly constructive. Here, we build on the trigonometric polynomial formulation to provide a streamlined proof of smoothness and, crucially, derive a bound on the curvature that highlights how circuit architecture and generator norms directly determine optimization geometry. Our main contributions are:

**A Formal Proof of L-Smoothness:** We leverage the established trigonometric polynomial structure of VQA objectives Wierichs et al. (2022) to provide a direct and intuitive proof of their L-smoothness. While this structure has been used for gradient analysis, we exploit it to derive a bound on the curvature. **A Tight and Explicit Upper Bound:** We derive a novel upper bound for the smoothness constant $L$ that is never looser and often tighter than previously known, with strict improvements when generator norms are heterogeneous or observables have non-commuting terms. The bound scales linearly with the number of parameters $P$ and is given by: $L \leq 4 \|M\|_2 \sum_{k=1}^{P} \|G_k\|_2^2$. For VQAs using standard single or two-qubit Pauli-rotation gates, this bound takes the remarkably simple form $L \leq P\|M\|_2$ and can sometimes be simplified further to $L \leq P$ for several observables. **Practical Landscape-Based Tools:** We empirically validate our bound, confirming that the landscape's maximum curvature scales in a predictable linear fashion with circuit depth. This stable geometric behavior yields two

Table 1: Comparison of VQA smoothness bounds. $P$ is the number of parameters, $G_k$ are the gate generators, and $\mu_k$ are the coefficients for observable $M$. Our bound is never looser, and can be up to a factor of $P$ tighter than Liu et al. (2025) and $\Theta(\sqrt{r})$ tighter than Gu et al. (2021).

| Bound | Assumptions / Setting | Tightness / Notes |
|---|---|---|
| Gu et al. Gu et al. (2021) $L \leq P \sum_k |\mu_k|$ | Gates must be Pauli rotations. | Often the loosest. Our bound is tighter by up to $\Theta(\sqrt{r})$ for observables with $r$ anti-commuting terms. |
| Liu et al. Liu et al. (2025) $L \leq 4\|M\|_2 P (\max_k \|G_k\|_2)^2$ | General gates (no Pauli-only restriction). | Our bound is tighter by up to a factor of $P$ (non-uniform generators). |
| **Ours** $L \leq 4\|M\|_2 \sum_{k=1}^{P} \|G_k\|_2^2$ | General gates (no Pauli-only restriction). | Correctly captures individual generator contributions and is general. |

powerful, practical applications. First, the predictable scaling allows us to introduce a computationally efficient heuristic for setting near-optimal learning rates, grounding this critical hyperparameter choice in landscape properties. Second, we show that the saturation point itself serves as a direct geometric signature of the circuit reaching its expressibility limit, offering a novel diagnostic for identifying inefficient overparameterization. **Validation in Noisy Environments:** Finally we demonstrate that our learning rate heuristic remains robust in NISQ environments, enabling Adam and SGD to achieve convergence rates competitive with the computationally expensive Quantum Natural Gradient (QNG) optimizer.

## 2 RELATED WORK

Although the optimization of VQAs is a central challenge, theoretical guarantees for the stability of gradient-based methods often rely on unproven smoothness assumptions or overly conservative bounds. Our work addresses these gaps by formally proving that VQA objective functions are globally L-smooth and by deriving a tight, computable expression for the smoothness constant. The concept of Lipschitz smoothness in VQAs is not entirely new: Gu et al. (2021) provided an observable-based upper bound used in adaptive shot allocation finding $L \leq P \sum |\mu_k|$, and Liu et al. (2025) introduced a "strong smoothness" property to analyze noisy optimization finding $L \propto P \cdot \max(\|G_k\|_2)^2$. However, both bounds are coarse. In contrast, we prove that the ideal VQA objective is globally smooth and derive an explicit, generator-aware bound. We show in Appendix C our bound is never looser, and often significantly tighter than both Gu et al. (2021) and Liu et al. (2025) bounds, in realistic Hamiltonians and non-uniform VQAs. Complementary to our work on inherent landscape geometry, other research has focused on bounding the statistical errors that arise when estimating derivatives from finite samples Teo (2023).

A key motivation for our work is that L-smoothness is a standard assumption in classical optimization, yet it is often assumed without proof in the VQA literature. For instance, convergence theories for over-parameterized VQEs You et al. (2022) and geometric methods like the Quantum Natural Gradient Stokes et al. (2020); Amari (1998) implicitly rely on a well-behaved landscape. More explicitly Koridon et al. (2024); Sweke et al. (2020); Gu et al. (2021) assume a Lipschitz-continuous Hessian. Our work replaces this common assumption with a provable guarantee, strengthening the theoretical foundations of these analyses. Furthermore, while other studies have provided valuable local characterizations of the Hessian Sen et al. (2022) or the related Fisher Information Matrix Abbas et al. (2021), our analysis establishes a formal and *global* property. We prove the Hessian norm is universally bounded, providing the theoretical guarantee that underpins these observations. Much foundational research on VQA landscapes has focused on first-order properties, driven by the need to mitigate the "barren plateau" phenomenon where gradients vanish McClean et al. (2018). This issue is deeply connected to ansatz expressibility, with highly expressive circuits being more prone to flat landscapes Sim et al. (2019); Holmes et al. (2022); Wang et al. (2021). These studies were driven from methods for calculating gradients, such as the parameter-shift rule Schuld et al. (2019), and deriving tight bounds on gradient concentrations Letcher et al. (2024). However these first-order characteristics are not sufficient for stable optimization. Our work provides the necessary second-order control by establishing an upper bound on landscape curvature, ensuring that the gradients obtained via these methods can be used effectively for training. This theoretical result also

informs practical VQA design; for instance, it provides a tool to quantify how circuit compression techniques Sim et al. (2019); Hu et al. (2022); Kulshrestha et al. (2024); Kverne et al. (2025) affect global landscape geometry. Finally, we distinguish our work, which proves smoothness with respect to circuit *parameters* for optimization stability, from research on the Lipschitz continuity of the objective function itself Sweke et al. (2020), a property that bounds the gradient's magnitude, or Lipschitz continuity with respect to *input data*, which addresses model robustness Berberich et al. (2024).

## 3 PRELIMINARIES

In this section, we formally define the components of a Variational Quantum Algorithm (VQA) and the concept of L-smoothness, establishing the notation used throughout our theoretical analysis.

### 3.1 VARIATIONAL QUANTUM CIRCUITS

A VQA operates on an $n$-qubit system, whose state is a vector in a $2^n$-dimensional Hilbert space $\mathcal{H}$ Nielsen & Chuang (2010). The algorithm begins with the system in a fixed initial state, typically the all-zeros state $|\psi_0\rangle = |0\rangle^{\otimes n}$ Peruzzo et al. (2014). A parameterized quantum circuit, or ansatz, is a unitary operator $U(\boldsymbol{\theta})$ that depends on a vector of $P$ real-valued parameters $\boldsymbol{\theta} \in \mathbb{R}^P$. This unitary transforms the initial state into a final parameterized state:

$$|\psi(\boldsymbol{\theta})\rangle = U(\boldsymbol{\theta}) |\psi_0\rangle \tag{1}$$

The ansatz $U(\boldsymbol{\theta})$ is constructed from a sequence of quantum gates. A common structure consists of $D$ layers, where each layer contains a set of parameterized gates followed by a set of fixed gates:

$$U(\boldsymbol{\theta}) = \prod_{d=D}^{1} U_d(\boldsymbol{\theta}_d)W_d \tag{2}$$

Here, $U_d(\boldsymbol{\theta}_d)$ is the unitary corresponding to the parameterized gates in layer $d$, and $W_d$ is the unitary for the fixed gates (e.g., CNOTs for entanglement).

### 3.2 PARAMETERIZED GATES AND GENERATORS

The parameterized gates are the core trainable components of the ansatz. Each gate $U_k(\theta_k)$ corresponding to a single parameter $\theta_k$ is generated by a Hermitian operator $G_k$ according to the Schrödinger equation:

$$U_k(\theta_k) = \exp(-i\theta_k G_k) \tag{3}$$

### 3.3 THE OBJECTIVE FUNCTION

The output of a VQA is the expectation value of a Hermitian operator $M$, known as an observable, with respect to the final state $|\psi(\boldsymbol{\theta})\rangle$. This expectation value serves as the objective function $f(\boldsymbol{\theta})$ that is minimized or maximized during training. For machine learning applications where a classical loss $\mathcal{C}(\bar{y}_{true}, f(\boldsymbol{\theta}))$ is used, our smoothness results for $f(\boldsymbol{\theta})$ extend directly, as standard loss functions are themselves smooth:

$$f(\boldsymbol{\theta}) \triangleq \langle\psi(\boldsymbol{\theta})|M|\psi(\boldsymbol{\theta})\rangle = \langle\psi_0| U^\dagger(\boldsymbol{\theta})MU(\boldsymbol{\theta}) |\psi_0\rangle \tag{4}$$

The observable $M$ is chosen based on the problem of interest. For example, in the Variational Quantum Eigensolver (VQE), $M$ is the Hamiltonian of a physical system whose ground state energy is sought.

### 3.4 DEFINITION: L-SMOOTHNESS

Our analysis centers on the smoothness of the objective function $f(\boldsymbol{\theta})$. A differentiable function is said to be L-smooth if its gradient is Lipschitz continuous.

**Definition 1.** *A function $f : \mathbb{R}^P \to \mathbb{R}$ is L-smooth for a constant $L \geq 0$ if $\forall \boldsymbol{\theta}_1, \boldsymbol{\theta}_2 \in \mathbb{R}^P$, the following inequality holds Nesterov (2013):*

$$\|\nabla f(\boldsymbol{\theta}_1) - \nabla f(\boldsymbol{\theta}_2)\|_2 \leq L \|\boldsymbol{\theta}_1 - \boldsymbol{\theta}_2\|_2 \tag{5}$$

*For a twice-differentiable function, a sufficient condition for L-smoothness is that the spectral norm of its Hessian matrix is globally bounded Nesterov (2013):*

$$\|\nabla^2 f(\boldsymbol{\theta})\|_2 \leq L \tag{6}$$

## 4 CHARACTERISTICS OF VQA FUNCTIONS

In this section, we present our core theoretical results. We first prove that the objective function of a VQA is a multivariate trigonometric polynomial (MTP), as shown in Lemma 1. This structural property is a key insight, as it means the function is a finite sum of sinusoids and therefore analytic and infinitely differentiable, which is the underlying reason its landscape is well-behaved. Building on this, we then derive an explicit upper bound on the L-smoothness constant by directly bounding the spectral norm of the Hessian matrix, formally proving that VQAs are L-smooth. Further details and a warm-up example are provided in Lemma 3 and Appendix B.

### 4.1 EXISTENCE OF A SMOOTHNESS BOUND

**Lemma 1.** *The objective function $f(\boldsymbol{\theta})$ of a VQA, as defined in Section 3, is a multivariate trigonometric polynomial of the circuit parameters $\boldsymbol{\theta}$.*

We show the proof of Lemma 1 in Appendix A.1. Because the objective function is a multivariate trigonometric polynomial, its derivatives have a structured Fourier form. This representation provides a route to establish smoothness: one can show that the Hessian operator norm is globally bounded in terms of Fourier amplitudes and frequencies. We give this derivation in Appendix A.6; while somewhat more technical, it yields the same upper bound as our generator-norm proof from Theorem 1. In the main text, we proceed with the generator-norm route, which provides a simpler and more direct path to the explicit smoothness constant.

### 4.2 FINDING AN EXPLICIT UPPER BOUND

**Theorem 1.** *The entries of the hessian matrix, $H = \nabla^2 f(\boldsymbol{\theta})$, are element-wise bounded by: $|H_{kl}| \leq 4\|M\|_2 \|G_k\|_2 \|G_l\|_2$*

*Proof.* We begin with the exact expression for the Hessian entries of the objective function $f(\boldsymbol{\theta}) = \langle \psi(\boldsymbol{\theta})| M |\psi(\boldsymbol{\theta})\rangle$ where $|\psi(\boldsymbol{\theta})\rangle = U(\boldsymbol{\theta}) |\psi_0\rangle$ and the unitary $U(\boldsymbol{\theta}) = \prod_{j=1}^{P} U_j(\theta_j) = U_P(\theta_P) \cdots U_k(\theta_k) \cdots U_1(\theta_1)$ is composed of gates $U_k(\theta_k) = \exp(-i\theta_k G_k)$. As derived in Appendix B Lemma 3, this is given by:

$$H_{kl} = \frac{\partial^2 f}{\partial \theta_l \partial \theta_k} = 2\mathrm{Re} \left( \frac{\partial \langle \psi(\boldsymbol{\theta})|}{\partial \theta_l} M \frac{\partial |\psi(\boldsymbol{\theta})\rangle}{\partial \theta_k} + \langle \psi(\boldsymbol{\theta})| M \frac{\partial^2 |\psi(\boldsymbol{\theta})\rangle}{\partial \theta_l \partial \theta_k} \right)$$

To bound $|H_{kl}|$ , we first establish bounds on the norms of the first and second derivatives of the statevector $|\psi(\boldsymbol{\theta})\rangle$. The first partial derivative with respect to $\theta_k$ is:

$$\frac{\partial |\psi(\boldsymbol{\theta})\rangle}{\partial \theta_k} = \left( \prod_{j=P}^{k+1} U_j(\theta_j) \right) (-iG_k U_k(\theta_k)) \left( \prod_{j=k-1}^{1} U_j(\theta_j) \right) |\psi_0\rangle$$

To bound its norm, we apply the submultiplicativity of the spectral norm ($\|AB\|_2 \leq \|A\|_2 \|B\|_2$). Since all $U_j$ are unitary operators ($\|U_j\|_2 = 1$) and the initial state is a unit vector ($\||\psi_0\rangle\|_2 = 1$), the expression simplifies:

$$\left\| \frac{\partial |\psi(\boldsymbol{\theta})\rangle}{\partial \theta_k} \right\|_2 \leq \left( \prod_{j=P}^{k+1} \|U_j\|_2 \right) \| -iG_k\|_2 \|U_k\|_2 \left( \prod_{j=k-1}^{1} \|U_j\|_2 \right) \||\psi_0\rangle\|_2 = \|G_k\|_2$$

The bound for the second derivative follows the same logic. For $l \neq k$ (assuming $l > k$ without loss of generality):

$$\left\|\frac{\partial^2 |\psi(\boldsymbol{\theta})\rangle}{\partial \theta_l \partial \theta_k}\right\|_2 = \left(\prod_{j=P}^{l+1} ||U_j||_2\right) ||-iG_l||_2 ||U_l||_2 \left(\prod_{j=l-1}^{k+1} ||U_j||_2\right) ||-iG_k||_2 ||U_k||_2 \left(\prod_{j=k-1}^{1} ||U_j||_2\right) |||\psi_0\rangle||_2$$

Taking the norm yields:

$$\left\|\frac{\partial^2 |\psi(\boldsymbol{\theta})\rangle}{\partial \theta_l \partial \theta_k}\right\|_2 \leq ||-iG_l||_2 \cdot ||-iG_k||_2 = ||G_l||_2 ||G_k||_2$$

For the unmixed case where $l = k$, the derivative contains a factor of $(-iG_k)^2$, leading to a norm bound of $||G_k^2||_2 \leq ||G_k||_2^2$. Thus, the general bound $||\frac{\partial^2 |\psi(\boldsymbol{\theta})\rangle}{\partial \theta_l \partial \theta_k}||_2 \leq ||G_l||_2 ||G_k||_2$ holds for all $k, l$.

Now we can bound the magnitude of each element $|H_{kl}|$ using the triangle inequality and the generalized Cauchy-Schwarz inequality ($|\langle u|A|v\rangle| \leq ||u||_2 ||A||_2 ||v||_2$):

$$|H_{kl}| \leq 2\left(\left|\frac{\partial \langle \psi(\boldsymbol{\theta})|}{\partial \theta_l} M \frac{\partial |\psi(\boldsymbol{\theta})\rangle}{\partial \theta_k}\right| + \left|\langle \psi(\boldsymbol{\theta})| M \frac{\partial^2 |\psi(\boldsymbol{\theta})\rangle}{\partial \theta_l \partial \theta_k}\right|\right)$$

$$\leq 2\left(\left\|\frac{\partial |\psi(\boldsymbol{\theta})\rangle}{\partial \theta_l}\right\|_2 ||M||_2 \left\|\frac{\partial |\psi(\boldsymbol{\theta})\rangle}{\partial \theta_k}\right\|_2 + ||\langle \psi(\boldsymbol{\theta})|||_2 ||M||_2 \left\|\frac{\partial^2 |\psi(\boldsymbol{\theta})\rangle}{\partial \theta_l \partial \theta_k}\right\|_2\right)$$

Applying the bounds for the second and first derivatives of the state vector and noting that $||\langle \psi(\boldsymbol{\theta})|||_2 = 1$:

$$|H_{kl}| \leq 2\left(||G_l||_2 ||M||_2 ||G_k||_2 + 1 \cdot ||M||_2 \cdot (||G_l||_2 ||G_k||_2)\right) = 4||M||_2 ||G_k||_2 ||G_l||_2$$

$$\square$$

Theorem 1 establishes that every entry of the Hessian matrix is globally bounded by a finite constant. This result is significant because a matrix with universally bounded entries necessarily has a bounded spectral norm. By definition 1, since a globally bounded Hessian norm is a sufficient condition for L-smoothness we have now formally proven that the VQA objective function is an $L$-smooth function. The remaining task is to find a tight, explicit value for the smoothness constant, $L$. The following theorem achieves this by deriving a closed-form upper bound on the Hessian's spectral norm.

**Lemma 2.** *Let $A, B \in \mathbb{R}^{n \times n}$. If $|A_{ij}| \leq B_{ij}$ for all $i, j$ and $B$ has non-negative entries, then $||A||_2 \leq ||B||_2$.*

*Proof.* For any vector $x \in \mathbb{R}^n$, $||Ax||_2 = ||\sum_j A_{ij} x_j||_2 \leq ||\sum_j |A_{ij}||x_j|||_2 \leq ||B|x|||_2$, where $|x|$ denotes the component wise absolute value. Taking the maximum over all unit vectors $x$ yields: $||A||_2 \leq \max_{||y||_2=1, y \geq 0} ||By||_2 \leq ||B||_2$. $\square$

**Theorem 2.** *The objective function $f(\boldsymbol{\theta})$ is L-smooth, with the smoothness constant L satisfying the upper bound:*

$$\boxed{L \leq 4||M||_2 \sum_{k=1}^{P} ||G_k||_2^2}$$

*Proof.* From Theorem 1, we have the element-wise inequality $|H_{kl}| \leq B_{kl}$, where $B$ is a matrix with non-negative entries $B_{kl} = 4||M||_2 ||G_k||_2 ||G_l||_2$. Following Lemma 2 we can denote that $||H||_2 \leq ||B||_2$. The matrix $B$ has a specific, low-rank structure. It can be expressed as the outer product of a vector with itself making it a positive semidefinite rank-1 matrix:

$$B = (4||M||_2) \cdot vv^T, \quad \text{where} \quad v = (||G_1||_2, \ldots, ||G_P||_2)^T$$

The spectral norm of a rank-1 matrix is equal to its single non-zero eigenvalue, which can be calculated directly. The norm is given by: $||B||_2 = (4||M||_2) \cdot ||v||_2^2$. The squared norm of the vector $v$ is simply

the sum of the squares of its components: $\|v\|_2^2 = \sum_{k=1}^{P}(\|G_k\|_2)^2 = \sum_{k=1}^{P}\|G_k\|_2^2$. Combining these steps, we arrive at the explicit bound for the spectral norm of $B$: $\|B\|_2 = 4\|M\|_2 \sum_{k=1}^{P}\|G_k\|_2^2$. Since $\|H\|_2 \leq L$ and we've shown $\|H\|_2 \leq \|B\|_2$, we have our final bound on the smoothness constant: $L \leq 4\|M\|_2 \sum_{k=1}^{P}\|G_k\|_2^2$ $\qquad\qquad\square$

This bound provides a clear theoretical prediction for the scaling of landscape curvature. It establishes that the maximum curvature grows at most linearly with the number of parameters $P$ and is scaled by both the observable norm ($\|M\|_2$) and the squared norms of the gate generators ($\|G_k\|_2^2$). This predicted linear scaling is a key feature of VQA landscapes, which we validate and analyze in detail in Section 5 (Fig. 1). A key advantage of our bound is its precision for circuits with non-uniform gate generators. By summing the individual squared norms, our formulation provides a provably tighter estimate than Liu et al. (2025) that scales with the maximum generator norm, as we confirm in Appendix C. The tightness of this bound is determined by the conditions under which the triangle and generalized Cauchy-Schwarz inequalities become equalities. While a gap between our theoretical limit and the true maximum curvature is inherent to this standard bounding technique, the crucial contribution is that our bound correctly captures the fundamental scaling relationships and provides the tightest known analytical estimate for general VQA landscapes.

**Corollary 1.** *For a VQA constructed from common gate sets and observables, the general smoothness bound simplifies significantly. Key special cases include:* **Pauli Rotation Gates:** *If the circuit uses only standard single-qubit or two-qubit (e.g, $R_x$, $R_{yy}$, etc.) Pauli rotation gates the bound becomes: $L \leq P\|M\|_2$.* **Pauli Gates and Normalized Observable:** *If the VQA uses the Pauli rotation gates above and the observable is normalized such that $\|M\|_2 = 1$, the bound simplifies further to: $L \leq P$.*

*Proof.* The proofs for these cases follow directly from Theorem 2.

The generator for a single-qubit Pauli rotation is $G_k = \frac{1}{2}P_i$ (where $P_i \in \{X, Y, Z\}$), and for a two-qubit rotation is $G_k = \frac{1}{2}P_i \otimes P_j$. Since $\|P_i\|_2 = 1$ and the spectral norm is submultiplicative over the tensor product, we have $\|G_k\|_2 = \frac{1}{2}$ in both cases. Substituting this into the general bound from Theorem 2 gives:

$$L \leq 4\|M\|_2 \sum_{k=1}^{P}\left(\frac{1}{2}\right)^2 = 4\|M\|_2 \sum_{k=1}^{P}\frac{1}{4} = P\|M\|_2$$

The condition $\|M\|_2 = 1$ is met in many common applications, such as when the observable is any Pauli operator (e.g., $M = Z_i$) or an average of Pauli operators like $M = \frac{1}{n}\sum_{i=0}^{n-1} Z_i$. With $\|M\|_2 = 1$, the bound $L \leq P\|M\|_2$ simplifies directly to $L \leq P$. $\qquad\square$

**Remark on Generality**: While our derivation focuses on standard single-parameter gates, the bound extends naturally to other architectures. For multi-parameter gates generated by a single Hermitian $G$, the bound remains valid using the spectral norm of $G$ (Corollary 1). In architectures with parameter sharing (e.g., QCNN Cong et al. (2019)), the objective retains its trigonometric polynomial structure (Appendix A.1). In these cases, the smoothness constant is bounded by the sum of the norms of all generators influenced by the shared parameter.

## 5 VALIDATING THE BOUND AND CONNECTING LANDSCAPE GEOMETRY TO EXPRESSIBILITY

In this section, we empirically validate our theoretical bound and investigate the geometry of VQA optimization landscapes. Our derived bound, $L \leq 4\|M\|_2 \sum_{k=1}^{P}\|G_k\|_2^2$, provides an analytical baseline that links the landscape's maximum curvature to its core components. We use this framework to systematically answer a fundamental question in VQA design: how do architectural choices (width $n$ vs. depth $P$), problem definition (observable norm $\|M\|_2$), and the choice of generators $G_k$ impact the landscape's curvature and, consequently, the feasibility of training? Furthermore, our analysis reveals a direct connection between landscape curvature and ansatz expressibility, establishing a geometric signature for the onset of inefficient overparameterization. Our analysis throughout this section focuses on the idealized properties of VQA landscapes. This allows us to establish a theoretical

baseline and understand the fundamental geometric contributions of circuit architecture, which is a prerequisite for analyzing more complex, noisy scenarios.

## 5.1 EXPERIMENTAL SETUP

Our setup consists of four numerical experiments to isolate the factors mentioned above. In all experiments, we estimated the true maximum curvature, $\tilde{L}_{max} = \max_{\boldsymbol{\theta} \in S} ||\nabla^2 f(\boldsymbol{\theta})||_2$, by sampling 1000 random parameter vectors, $S = \{\boldsymbol{\theta}_i\}_{i=1}^{1000}$, and reporting the largest observed Hessian norm, $\tilde{L}_{max}$. While computing the global maximum curvature is provably intractable Bittel & Kliesch (2021), we found a sample set of $S = 1000$ to be sufficient for a stable estimate of the maximum curvature while being computationally feasible. As detailed in Appendix D.2, our convergence analysis shows the estimate stabilizes rapidly, and independent runs yield low variance, confirming this method captures the characteristic geometry of the landscape without the prohibitive cost of global optimization. Our baseline ansatz consists of $D$ layers of single-qubit rotations ($R_x, R_y, R_z$) on each of the $n$ qubits, followed by a cycle of CNOT gates for entanglement. Our analysis centers on the ratio of the empirical curvature to our theoretical upper bound, $\tilde{L}_{max}/L_{upper}$, to quantify the bound's tightness and understand the landscape's scaling properties. For further details see Appendix D. **Experiment 1 (Depth and Width):** To isolate the effect of $P$, we used standard Pauli rotation gates ($||G_k||_2 = 1/2$) and a normalized observable $M = \frac{1}{n}\sum_i Z_i$ ($||M||_2 = 1$), simplifying our bound to $L_{upper} = P$. We varied the qubit count $n \in \{1, 2, 4, 8, 10\}$ and depth up to $P = 120$. This experiment was repeated for different VQA architectures in Appendix D.3. **Experiment 2 (Observable Norm):** To test the linear scaling with $||M||_2$, we fixed the architecture ($n \in \{2, 4, 8\}, D = 2$) and used the observable $M = Z_0 \otimes Z_1 + w(X_0 + X_1)$, varying $w$ from 0.1 to 5.0 to monotonically increase $||M||_2$. This observable represents the Hamiltonian for the transverse-field Ising model, a classic problem in condensed matter physics and a standard benchmark for algorithms like VQEs Peruzzo et al. (2014).

**Experiment 3 (Expressibility):** Third, we investigate the link between landscape geometry and the ansatz's "expressibility". A highly expressive model can uniformly explore the entire Hilbert space, which is crucial for its potential to solve a wide range of problems. We quantify this using the method from Sim et al. (2019), which compares the model's output distribution to a theoretical ideal using KL divergence. First, we create an empirical distribution by sampling pairs of random parameter vectors $(\boldsymbol{\theta}_1, \boldsymbol{\theta}_2)$, generating the corresponding states ($|\psi(\boldsymbol{\theta}_1)\rangle, |\psi(\boldsymbol{\theta}_2)\rangle$), and calculating their similarity, or fidelity, $F = |\langle\psi(\boldsymbol{\theta}_1)|\psi(\boldsymbol{\theta}_2)\rangle|^2$. This distribution is then compared to the theoretical fidelity distribution of "Haar-random" states, which represent a perfectly uniform sampling of the state space. A low KL divergence ($D_{KL}$) signifies that the ansatz is not confined to a small corner of the state space but can generate outputs that are representative of the space as a whole, making it highly expressive. This metric is known to exhibit "expressibility saturation": as circuit depth increases, expressivity improves (lower $D_{KL}$), but eventually plateaus. At this point, adding more parameters yields diminishing returns in representative power, a direct analogue to inefficient overparameterization in classical models Larocca et al. (2023); You et al. (2022). We extended Experiment 1 to test whether this saturation point has a direct geometric signature in our landscape curvature metric, allowing for a side-by-side comparison of the landscape's geometry with the circuit's state-space coverage. Further details are in Appendix D.1.1.

**Experiment 4 (Generator Norms):** To validate the tightness and practical advantages of our bound, we conducted two distinct numerical experiments comparing it against prior work. Comparison with Liu et al. (2025): We test our general bound in scenarios with non-uniform generators. We fixed the architecture ($D = 2, n = \{2, 4, 8\}$) and observable ($||M||_2 = 1$). We then constructed circuits by mixing standard Pauli rotations ($||G_k||_2^2 = 1/4$) and weighted rotations where $||G_k^W|| = 4$. we varied the ratio of Pauli and weighted gates between 0% and 100%. Comparison with Gu et al. (2021): We compared our simplified bound $L \leq P||M||_2$ for circuits with standard Pauli rotations for physically-motivated Hamiltonians. We used the same fixed ansatz ($D = 2, n = \{2, 4, 8\}$) and measure the landscape curvature for three different observables being the Ising Model (non-commuting Pauli X and Z terms), Heisenberg Model (non-commuting XX, YY, and ZZ terms), and Mixed-Field Model (more complex with several non-commuting terms) Results are shown in Appendix C.

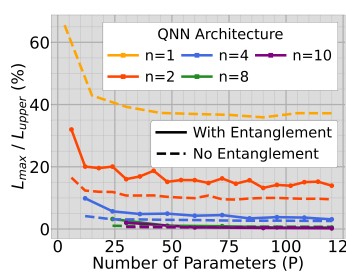
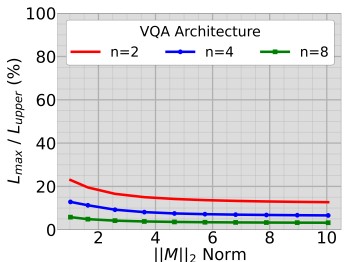
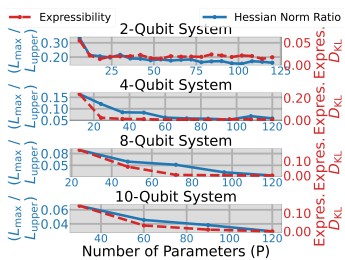

(a) ($\tilde{L}_{max}$ / $L_{upper}$) Scaling with number of parameters (P).

(b) ($\tilde{L}_{max}$ / $L_{upper}$) Scaling with observable norm ($\|M\|_2$).

(c) Direct comparison of landscape curvature and expressibility.

Figure 1: Empirical validation of the smoothness bound and demonstration of the link between landscape curvature and expressibility saturation.

### 5.1.1 FINDING 1: CURVATURE VANISHES WITH WIDTH — A GEOMETRIC VIEW OF BARREN PLATEAUS

As shown in Figure 1a and 1b, when the number of qubits $n$ increases while depth is held constant, the ratio $\tilde{L}_{max}/L_{upper}$ decreases rapidly. This demonstrates that the true landscape curvature collapses far more rapidly than our bound, which is linear in $P$. This finding provides a new, second-order signature of the barren plateau phenomenon. While prior work focused on vanishing gradients (a first-order effect), our results show that as the Hilbert space grows, the landscape undergoes a geometric phase transition into a flat regime where second-order information also vanishes, making gradient-based optimization challenging to use with more qubits.

### 5.2 FINDING 2: PREDICTABLE CURVATURE SCALING AND ITS LINK TO OVERPARAMETERIZATION

In stark contrast, our experiments focused on depth and observable norm reveal a stable and predictable landscape geometry. As shown in Figure 1a, for a fixed number of qubits $n$, increasing the number of parameters $P$ (i.e., depth) causes the ratio $\tilde{L}_{max}/L_{upper}$ to stabilize to a constant, non-zero value. We observe the exact same stabilization behavior in Figure 1b, where we fix the circuit architecture and increase the observable's norm, $\|M\|_2$. This stabilization is a crucial finding as the constant ratio implies that the true maximum curvature scales linearly with both the number of parameters and the observable's norm. In other words, the true maximum curvature obeys:

$$\tilde{L}_{max} \propto P \quad \text{and} \quad \tilde{L}_{max} \propto \|M\|_2$$

This predictable linear scaling is a key theoretical finding. It demonstrates that landscape curvature grows in a controlled manner with circuit depth, rather than becoming unstable or chaotic. This regularity is crucial, as it ensures the robustness of gradient-based optimizers and establishes that VQA landscapes possess a far more manageable structure than worst-case analyses might suggest. As shown in Figure 1c, as we add parameters to the circuit, the landscape curvature ratio ($\tilde{L}_{max}/L_{upper}$) initially decreases rapidly. This happens in lockstep with the expressibility metric ($D_{KL}$), which also decreases as the circuit becomes more capable of exploring the Hilbert space. However, as more parameters are added, both curves eventually begin to plateau (we observe the plateau effect for a wider range of architectures in Appendix D.3). This indicates that the circuit has hit its "expressibility limit" for a given width, where adding more layers provides diminishing returns. This observation aligns with recent theoretical work on overparameterization based on Dynamical Lie Algebras Larocca et al. (2023), providing a complementary, landscape-based signature of when the ansatz becomes inefficient. The patterns of initial decrease followed by saturation are remarkably similar for both our geometric metric and the expressibility metric. We theorize that the underlying reason for this strong correlation lies in the concept of parameter redundancy. When a circuit is underparameterized, each parameter is largely non-redundant, providing access to new regions of the Hilbert space and thus having a unique and strong influence on the objective function's gradient, which leads to a landscape with high maximum curvature. Conversely, in the saturated regime, parameters become

increasingly redundant as the transformations from new gates can be approximated by combinations of existing ones. As this functional overlap increases, the landscape's curvature stops growing relative to its theoretical bound, causing the $\tilde{L}_{max}/L_{upper}$ ratio to flatten out precisely when the circuit's expressibility saturates. Therefore, the plateau in the $\tilde{L}_{max}/L_{upper}$ ratio is the geometric manifestation of the circuit reaching the limit of its representative power for a given width. This establishes our curvature ratio as a powerful and practical diagnostic tool. It can signal the onset of inefficient overparameterization directly from the properties of the optimization landscape, providing a complementary and computationally accessible method for guiding ansatz design.

## 5.3 HEURISITC FOR SETTING NEAR OPTIMAL LEARNING RATES

The optimal learning rate, $\eta^*_{local}$, at any point in the landscape is inversely proportional to the local curvature, $L_{\boldsymbol{\theta}} = ||\nabla^2 f(\boldsymbol{\theta})||_2$. Since computing this local value at every step is intractable, a common strategy is to use a single global learning rate, $\eta_{global}$, chosen to be stable even in the regions of highest curvature. This choice is ideally based on the global maximum curvature, $\tilde{L}_{max}$. Our findings, however, show that $\tilde{L}_{max}$ scales linearly with our bound allowing us to form a much tighter estimate of the true maximum curvature by defining an *effective smoothness constant*:

$$L_{\text{effective}} \triangleq \kappa(n) \cdot L_{\text{upper}}$$

where $\kappa(n)$ is a scaling factor. The stability in $\tilde{L}_{max}/L_{upper}$ for a given width ($n$) enables an efficient calibration procedure. Let the target circuit have a depth $D$ with a corresponding parameter vector $\boldsymbol{\theta} \in \mathbb{R}^{P(D)}$. A practitioner can estimate $\kappa(n)$ using a computationally inexpensive, shallow circuit of depth $D_{\text{cal}} \ll D$ with its own parameter vector $\boldsymbol{\theta}_{\text{cal}} \in \mathbb{R}^{P(D_{\text{cal}})}$:

$$\kappa(n) \approx \frac{\max_{\boldsymbol{\theta}_{\text{cal}}} ||\nabla^2 f(\boldsymbol{\theta}_{\text{cal}})||_2}{L_{\text{upper}}(P(D_{\text{cal}}))}$$

This pre-computed factor can then be used to set a near-optimal global learning rate for the much deeper target circuit of depth $D$:

$$\eta^*_{\text{global}} \approx \frac{1}{L_{\text{effective}}} = \frac{1}{\kappa(n) \cdot L_{\text{upper}}(P(D))}$$

This method replaces blind hyperparameter searches with a theoretically-grounded and empirically-calibrated strategy, directly connecting the learning rate to the circuit's fundamental geometric properties. We show that this method has a quadratic advantage with minimal error in Appendix 5.3 and find that the "plateau" effect can be observed in a broader range of VQA architectures in Appendix D.3 making this method general. Crucially, this heuristic is compatible with adaptive methods. The calculated $\eta^*_{global}$ serves as the optimal base learning rate for optimizers like Adam. To demonstrate the practical value of our heuristic, we applied it to a common VQA task: finding the ground state energy of the transverse-field Ising model using a Variational Quantum Eigensolver (VQE). For 1, 2, and 4-qubit systems, we first calibrated the scaling factor $\kappa(n)$ on a shallow circuit. We then used the resulting "optimal" learning rate to train much deeper VQEs with both SGD and ADAM Kingma & Ba (2014) optimizers. We compared its performance against three other learning rates: a high rate ($5 \times \eta^*_{global}$), a low rate ($0.2 \times \eta^*_{global}$), and a standard default rate (0.01 for SGD, 0.001 for ADAM). As shown in Figures 9 and 10 and summarized in Table 2 (Appendix 5.3), the calibrated learning rate consistently enables faster and more stable convergence than the other learning rates (on average the low learning rate converged 21% slower than the optimal one for ADAM, the high one was on average 54% slower for ADAM and the standard one was 582% slower for ADAM). Additional experimental details are given in Appendix F.2 with detailed results shown in Appendix F.2.1.

## 5.4 VALIDATION IN NOISY ENVIRONMENTS AND COMPARISON WITH QNG

Finally, to validate our bound in realistic settings, we extended the learning rate heuristic experiment to a noisy environment by inducing depolarizing noise channels after every single and double-qubit

gate, coupled with shot-based measurements (1000 shots). We applied our heuristic in this setting and compared it against the Quantum Natural Gradient (QNG) optimizer to assess its relative value. While QNG explicitly accounts for geometry by computing the Fubini-Study metric tensor at every step, this operation is computationally expensive ($\mathcal{O}(P^2)$). As shown in our results in Appendix F (Figure 11, Table 3), our heuristic enables significantly faster convergence while reaching lower energies compared to standard baselines even in this noisy regime, signaling that our bound accurately captures the landscape curvature despite the noise. Furthermore, the calibrated learning rate yields more stable convergence (lower standard deviation) than uncalibrated variants and achieves performance competitive with the far more expensive QNG optimizer, offering a highly efficient alternative for NISQ training.

## 6 DISCUSSION

In this work, we formally established that VQA objective functions are inherently L-smooth and derived a tight upper bound for the smoothness constant, $L \leq 4\|M\|_2 \sum_{k=1}^{P} \|G_k\|_2^2$, which outperforms previous estimates. Our empirical analysis revealed that landscape curvature scales predictably and linearly with circuit depth. We demonstrate that the saturation of this scaling provides a direct, geometric signature of expressibility saturation, offering a powerful diagnostic tool to identify the onset of inefficient overparameterization. Additionally we utilize this plateau to develop a heuristic for setting near optimal learning rates for deep circuits. Our analysis is conducted in a, noiseless setting or artificially induced noisy environments, allowing for a precise study of the fundamental geometric properties of VQA landscapes. This approach is intentional and serves as a theoretical baseline. Future work should investigate how our smoothness bounds and the observed scaling phenomena behave on real quantum hardware. Specifically, a formal analytical derivation of the plateau factor $\kappa(n)$ remains an open theoretical challenge, likely requiring random matrix theory and integration over the Haar measure. Additionally, while our work establishes an upper bound on the Hessian, investigating lower bounds would be valuable for establishing Polyak-Lojasiewicz (PL) conditions to guarantee global convergence. Finally, exploring how this calibration integrates with parameter-free methods (e.g., COCOB) presents a promising direction for fully automated quantum optimization. Our estimates of the maximum curvature are obtained by random sampling of parameter vectors rather than solving the global maximization problem $\max_{\boldsymbol{\theta}} \|\nabla^2 f(\boldsymbol{\theta})\|_2$ While this stochastic approach cannot guarantee identification of the true maximum, our convergence experiments (Appendix D.2) demonstrate that a sample size of 1000 is sufficient for the estimates to stabilize with minimal variation, indicating that the values we report are reliable and representative of curvature. Importantly this method is computationally feasible as large-scale QML experiments are notoriously expensive due to the derivative computation methods (i.e. no backpropagation) and exponentially large Hilbert spaces they operate in opposed to the classical linear growth of neural networks.

## REPRODUCIBILITY STATEMENT

We are committed to ensuring the reproducibility of our work. All theoretical claims are formally proven in the main text, with detailed derivations presented in Section 4. Further justifications, including a step-by-step example of the trigonometric polynomial form and an expanded derivation of the Hessian, are provided in Appendices A and B. The setup for all numerical experiments is detailed in Section 5, with specific implementation details, software versions (PennyLane v0.38.0), and the methodology for quantifying expressibility further elaborated in Appendix D. The complete source code required to replicate our experiments, validate our bounds, and generate all figures is included in the supplementary materials.

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

# Appendices

## A  JUSTIFICATION FOR THE TRIGONOMETRIC POLYNOMIAL FORM

In the main text, we show that VQAs belong to a well behaved continuous class of functions being multivariate trigonometric polynomial. A critical step in this induction relies on the fact that the matrix elements of a parameterized unitary gate, $U_k(\theta_k)$, are themselves trigonometric polynomials of the parameter $\theta_k$. Here, we provide additional details for this property and show the proof for Lemma 1.

### A.1  PROOF OF LEMMA 1

*Proof.* We prove by induction on the number of applied gates, $k$, that each amplitude $c_i(\boldsymbol{\theta})$ of the state vector $|\psi(\boldsymbol{\theta})\rangle$ is a multivariate trigonometric polynomial. A function is a multivariate trigonometric polynomial if it can be expressed as a finite linear combination of complex exponentials. Mathematically, this means each amplitude $c_i(\boldsymbol{\theta})$ can be written in the form:

$$c_i(\boldsymbol{\theta}) = \sum_{\omega \in \Omega_i} d_\omega^{(i)} e^{i(\omega \cdot \boldsymbol{\theta})}$$

where: $\Omega_i$ is a finite set of real-valued frequency vectors $\omega = (\omega_1 \ldots, \omega_P)$, and $d_\omega^{(i)}$ are complex coefficients.

**Inductive Hypothesis:** Assume that after $k-1$ gates, the amplitudes of the state $|\psi_{k-1}\rangle$ are trigonometric polynomials of the parameters applied so far.

**Base Case ($k = 0$):** The circuit has no gates. The state is the initial state $|\psi_0\rangle = |0\rangle^{\otimes n}$, whose amplitudes are constants ($c_0 = 1$, and $c_{i>0} = 0$). Constants are trivial trigonometric polynomials of degree zero.

**Inductive Step:** Consider the $k$-th gate, $U_k$. The new state is $|\psi_k\rangle = U_k |\psi_{k-1}\rangle$, and its amplitudes $c_i'$ are given by the linear transformation $c_i' = \sum_j (U_k)_{ij} c_j$. We examine two cases for $U_k$:

**Case 1. $U_k$ is a parameterized gate:** Consider the k-th gate, $U_k(\theta_k) = \exp(-i\theta_k G_k)$. By the spectral theorem (see Appendix A.3), the matrix elements $(U_k)_{ij}$ are finite sums of terms like $a_{lm} e^{-i\theta_k \lambda_m}$, where $\lambda_m$ are the real eigenvalues of the generator $G_k$. Since the set of multivariate trigonometric polynomials is closed under addition and multiplication, the new amplitudes, which are linear combinations of the old amplitudes and the elements $(U_k)_{ij}$, are also multivariate trigonometric polynomials. The frequencies of the resulting polynomial are sums of the frequencies from previous steps and the eigenvalues of the generators applied.

**Case 2. $U_k$ is a fixed gate:** The matrix elements $(U_k)_{ij}$ are constant complex numbers. The new amplitudes are finite linear combinations of the previous amplitudes. By the inductive hypothesis, the previous amplitudes are trigonometric polynomials, and a linear combination of trigonometric polynomials is itself a trigonometric polynomial.

By the principle of induction, the amplitudes $c_i(\boldsymbol{\theta})$ of the final state vector are multivariate trigonometric polynomials. Finally, the objective function is given by $f(\boldsymbol{\theta}) = \sum_{i,j} c_i^*(\boldsymbol{\theta}) M_{ij} c_j(\boldsymbol{\theta})$. Since the complex conjugate of a trigonometric polynomial is also a trigonometric polynomial, $f(\boldsymbol{\theta})$ is a sum of products of trigonometric polynomials and constants ($M_{ij}$). It is therefore itself a multivariate trigonometric polynomial. $\qquad\square$

### A.2  WARM UP EXAMPLE FOR QUANTUM STATE EVOLUTIONS

To demonstrate the trigonometric polynomial property of VQAs (Lemma 1) let's begin our warm up example with a simple 2-qubit case initialized in:

$$|\psi_0\rangle = |0\rangle^{\otimes 2} = \begin{pmatrix} c_1 \\ c_2 \\ c_3 \\ c_4 \end{pmatrix} = \begin{pmatrix} 1 \\ 0 \\ 0 \\ 0 \end{pmatrix}$$

as our basecase follows each amplitude $c_1, c_2, c_3, c_4$ is a trigonometric polynomial of an arbitrary degree (i.e. $\sin^0(x)$ or $0 \cdot \sin(x)$). **Step 1: Applying a parametrized gate**. Let $U_1(\theta_1) = e^{-i\theta_1 X} = R_x(\theta_1)$. Its matrix form is:

$$R_x(\theta_1) = \begin{pmatrix} \cos(\theta_1/2) & -i\sin(\theta_1/2) \\ -i\sin(\theta_1/2) & \cos(\theta_1/2) \end{pmatrix}$$

To apply this gate to the first qubit of our 2-qubit system, we must apply the identity operator I to the second qubit. The full operator for the system is the tensor product of the two, which results in a 4×4 matrix:

$$U_{full}(\theta_1) = R_x(\theta_1) \otimes I = \begin{pmatrix} \cos(\theta_1/2) & -i\sin(\theta_1/2) \\ -i\sin(\theta_1/2) & \cos(\theta_1/2) \end{pmatrix} \otimes \begin{pmatrix} 1 & 0 \\ 0 & 1 \end{pmatrix}$$

$$= \begin{pmatrix} \cos(\theta_1/2) & 0 & -i\sin(\theta_1/2) & 0 \\ 0 & \cos(\theta_1/2) & 0 & -i\sin(\theta_1/2) \\ -i\sin(\theta_1/2) & 0 & \cos(\theta_1/2) & 0 \\ 0 & -i\sin(\theta_1/2) & 0 & \cos(\theta_1/2) \end{pmatrix}$$

The new state $|\psi_1\rangle$ is then given by applying this operator to the initial state $|\psi_0\rangle$:

$$|\psi_1\rangle = U_{full}(\theta_1)|\psi_0\rangle = \begin{pmatrix} \cos(\theta_1/2) & 0 & -i\sin(\theta_1/2) & 0 \\ 0 & \cos(\theta_1/2) & 0 & -i\sin(\theta_1/2) \\ -i\sin(\theta_1/2) & 0 & \cos(\theta_1/2) & 0 \\ 0 & -i\sin(\theta_1/2) & 0 & \cos(\theta_1/2) \end{pmatrix} \cdot \begin{pmatrix} 1 \\ 0 \\ 0 \\ 0 \end{pmatrix}$$

$$= \begin{pmatrix} \cos(\theta_1/2) \\ 0 \\ -i\sin(\theta_1/2) \\ 0 \end{pmatrix}$$

We can see that the new amplitudes of $|\psi_1\rangle$ are $\{\cos(\theta_1/2), 0, -i\sin(\theta_1/2), 0\}$. Each of these is a trigonometric polynomial of the parameter $\theta_1$, which is consistent with the inductive step for a parameterized gate

**Step 2: Applying a Fixed Gate**. Next, let's apply a fixed (non-parameterized) entangling gate, such as a CNOT gate where the first qubit is the control and the second is the target. The matrix for this gate is:

$$CNOT = \begin{pmatrix} 1 & 0 & 0 & 0 \\ 0 & 1 & 0 & 0 \\ 0 & 0 & 0 & 1 \\ 0 & 0 & 1 & 0 \end{pmatrix}$$

We apply this to our current state $|\psi_1\rangle$ to get the final state $|\psi_2\rangle$:

$$|\psi_2\rangle = CNOT|\psi_1\rangle = \begin{pmatrix} 1 & 0 & 0 & 0 \\ 0 & 1 & 0 & 0 \\ 0 & 0 & 0 & 1 \\ 0 & 0 & 1 & 0 \end{pmatrix} \begin{pmatrix} \cos(\theta_1/2) \\ 0 \\ -i\sin(\theta_1/2) \\ 0 \end{pmatrix} = \begin{pmatrix} \cos(\theta_1/2) \\ 0 \\ 0 \\ -i\sin(\theta_1/2) \end{pmatrix}$$

The final amplitudes are linear combinations of the previous amplitudes. Since the previous amplitudes were trigonometric polynomials, the new amplitudes are also trigonometric polynomials. This example demonstrates both cases of the inductive proof and shows how the amplitudes of the state vector maintain their trigonometric polynomial form throughout the circuit's evolution.

Finally let's measure the system by applying an observation $M = Z_0$. The matrix is given by the tensor product of the Pauli Z matrix on the first qubit and the Identity on the second: $Z \otimes I$:

$$M = \begin{pmatrix} 1 & 0 \\ 0 & -1 \end{pmatrix} \otimes \begin{pmatrix} 1 & 0 \\ 0 & 1 \end{pmatrix} = \begin{pmatrix} 1 & 0 & 0 & 0 \\ 0 & 1 & 0 & 0 \\ 0 & 0 & -1 & 0 \\ 0 & 0 & 0 & -1 \end{pmatrix}$$

Then we compute $\langle\psi_2| M |\psi_2\rangle$ which is:

$$
(\cos(\theta_1/2) \quad 0 \quad i\sin(\theta_1/2) \quad 0)
\begin{pmatrix}
1 & 0 & 0 & 0 \\
0 & 1 & 0 & 0 \\
0 & 0 & -1 & 0 \\
0 & 0 & 0 & -1
\end{pmatrix}
\begin{pmatrix}
\cos(\theta_1/2) \\
0 \\
0 \\
-i\sin(\theta_1/2)
\end{pmatrix}
$$

Computing this gives $\langle\psi_2| M |\psi_2\rangle = \cos^2(\theta_1/2) - \sin^2(\theta_1/2) = \cos(\theta_1)$, making the final term for the objective function is still a trigonometric polynomial.

## A.3 THE SPECTRAL THEOREM

The foundation for the proof of Lemma 1 is the spectral theorem, which states that any Hermitian operator $G_k$ can be diagonalized Nielsen & Chuang (2010). We can express $G_k$ as:

$$G_k = VDV^\dagger \tag{7}$$

where:

- $D$ is a real diagonal matrix containing the eigenvalues $(\lambda_1, \lambda_2, \ldots, \lambda_N)$ of $G_k$.
- $V$ is a unitary matrix whose columns are the corresponding eigenvectors of $G_k$.
- $V^\dagger$ is the conjugate transpose of $V$.

A key property of this decomposition is that applying a function $f$ to the matrix $G_k$ is equivalent to applying the function to its eigenvalues in the diagonal matrix $D$:

$$f(G_k) = Vf(D)V^\dagger \tag{8}$$

## A.4 DERIVING THE FORM OF A PARAMETERIZED GATE

The parameterized gate is defined by the matrix exponential function $f(x) = e^{-i\theta_k x}$ applied to $G_k$ it. Using the property from the spectral theorem, we can write the gate $U_k(\theta_k)$ as:

$$U_k(\theta_k) = \exp(-i\theta_k G_k) = V\exp(-i\theta_k D)V^\dagger \tag{9}$$

Since $D$ is a diagonal matrix, its exponential is simply the exponential of each of its diagonal elements:

$$
\exp(-i\theta_k D) =
\begin{pmatrix}
e^{-i\theta_k\lambda_1} & 0 & \cdots & 0 \\
0 & e^{-i\theta_k\lambda_2} & \cdots & 0 \\
\vdots & \vdots & \ddots & \vdots \\
0 & 0 & \cdots & e^{-i\theta_k\lambda_N}
\end{pmatrix}
\tag{10}
$$

## A.5 THE FORM OF A SINGLE MATRIX ELEMENT

To find the form of an individual entry $(U_k)_{ij}$ of the gate, we write out the full matrix multiplication $U_k = V\exp(-i\theta_k D)V^\dagger$. The entry at the $i$-th row and $j$-th column is given by:

$$(U_k)_{ij} = \sum_{m=1}^{N} V_{im} \left(\exp(-i\theta_k D)\right)_{mm} (V^\dagger)_{mj} \tag{11}$$

Substituting the diagonal elements of the exponential matrix, we get:

$$(U_k)_{ij} = \sum_{m=1}^{N} V_{im} \left(e^{-i\theta_k\lambda_m}\right) (V^\dagger)_{mj} \tag{12}$$

Let's analyze this expression. For a given gate, the eigenvector matrices $V$ and $V^\dagger$ are constant. Therefore, the terms $V_{im}$ and $(V^\dagger)_{mj}$ are simply complex coefficients. The only dependence on the parameter $\theta_k$ is contained within the complex exponential terms $e^{-i\theta_k\lambda_m}$. This shows that any matrix element $(U_k)_{ij}$ is a finite linear combination of complex exponentials of $\theta_k$. This is precisely the definition of a trigonometric polynomial. By Euler's formula, $e^{ix} = \cos(x) + i\sin(x)$, this sum can be equivalently expressed as a finite sum of sine and cosine functions of $\theta_k$, which completes the justification required for the inductive proof in the main text.

## A.6 PROVING L-SMOOTHNESS BY ANALYZING TRIGONOMETRIC POLYNOMIALS

As proven in Lemma 1 the objective function is a multivariate trigonometric polynomial and can therefore be written as a finite Fourier series:

$$f(\boldsymbol{\theta}) = \sum_{\omega \in \Omega} d_\omega e^{i\omega\boldsymbol{\theta}}$$

We find the hessian entries by differentiating twice:

$$H_{kl} = \frac{\partial^2}{\partial\theta_k \partial\theta_l} \sum_{\omega \in \Omega} d_\omega e^{i\omega\cdot\boldsymbol{\theta}} = \sum_{\omega \in \Omega} (i\omega_k)(i\omega_l) d_\omega e^{i\omega\cdot\boldsymbol{\theta}}$$

This simplifies to:

$$H_{kl} = - \sum_{\omega \in \Omega} \omega_k \omega_l d_\omega e^{i\omega\cdot\boldsymbol{\theta}}$$

Now, we take the magnitude of a single Hessian entry and apply the triangle inequality:

$$|H_{kl}| = \left| - \sum_{\omega \in \Omega} \omega_k \omega_l d_\omega e^{i\omega\cdot\theta} \right| \leq \sum_{\omega \in \Omega} |\omega_k \omega_l d_\omega e^{i\omega\cdot\theta}|$$

This can be broken down further:

$$|H_{kl}| \leq \sum_{\omega \in \Omega} |\omega_k||\omega_l||d_\omega||e^{i\omega\cdot\theta}|$$

Since $|e^{i\omega\cdot\theta}| = 1$, we get our final bound for a single element:

$$|H_{kl}| \leq \sum_{\omega \in \Omega} |\omega_k||\omega_l||d_\omega|$$

Bounding the frequencies: In the expansion of the expectation value $f(\boldsymbol{\theta}) = \langle\psi(\boldsymbol{\theta})|M|\psi(\boldsymbol{\theta})\rangle$, the Fourier frequencies along coordinate $k$ arise from *differences* of eigenvalues of the Hermitian generator $G_k$ (bra vs. ket paths). Hence

$$|\omega_k| \leq |\lambda| + |\lambda'| \leq 2\|G_k\|_2, \qquad |\omega_l| \leq 2\|G_l\|_2,$$

so that

$$|\omega_k \omega_l| \leq 4\|G_k\|_2\|G_l\|_2.$$

Fourier coefficients: Each Fourier coefficient can be written as a matrix element of $M$ between unit vectors generated by (products of) unitaries from the circuit; i.e., there exist unit vectors $(|u_\omega\rangle, |v_\omega\rangle)$ such that:

$$d_\omega = \langle u_\omega| M |v_\omega\rangle$$

By the Cauchy–Schwarz inequality, we obtain the uniform bound

$$|d_\omega| \leq \|M\|_2.$$

**Putting it together.** Substituting these bounds gives

$$|H_{kl}| \leq \sum_{\omega \in \Omega} |\omega_k| \, |\omega_l| \, |d_\omega| \; \leq \; \sum_{\omega \in \Omega} \left( 2\|G_k\|_2 \right) \left( 2\|G_l\|_2 \right) |d_\omega|$$

$$\leq \; 4 \, \|M\|_2 \, \|G_k\|_2 \, \|G_l\|_2.$$

Therefore, elementwise

$$|H_{kl}| \; \leq \; B_{kl} \quad \text{with} \quad B_{kl} \; = \; 4\|M\|_2 \, \|G_k\|_2 \, \|G_l\|_2 \,.$$

As in the main text Theorem 2, defining $v \in \mathbb{R}^P$ by $v_k = \|G_k\|_2$ yields $B = 4\|M\|_2 \, vv^T$, so $\|H\|_2 \leq \|B\|_2 = 4 \, \|M\|_2 \sum_{k=1}^P \|G_k\|_2^2$, recovering the bound in Theorem 2. This demonstrates that each entry $H_{kl}$ is bounded by a constant value. A matrix whose entries are all globally bounded has a bounded spectral norm. This is a sufficient condition to prove that the VQA objective function is L-smooth using the trigonometric polynomial form of VQAs.

## B    BOUNDING THE HESSIAN

Following Theorem 1 after reaching the inequality $|H_{kl}| \leq 4\|M\|_2\|G_k\|_2\|G_l\|_2$ we bound $L$ by finding its spectral norm $L \leq \|B\|_2$. It is worth noting that one could also arrive at this same result by bounding the spectral norm with the Frobenius norm ($\| \cdot \|_2 \leq \| \cdot \|_F$). While this inequality can be loose, in our specific case the bound matrix $B$ is a rank-1 matrix, for which the spectral and Frobenius norms are identical. Thus, both proof strategies are equally tight and yield the same bound.

### B.1    FINDING A TERM FOR THE HESSIAN ELEMENTS – DETAILED DERIVATION

The VQA objective function is defined as the expectation value of an observable $M$ with respect to the parameterized state $|\psi(\boldsymbol{\theta})\rangle$:

$$f(\boldsymbol{\theta}) = \langle \psi(\boldsymbol{\theta})| \, M \, |\psi(\boldsymbol{\theta})\rangle$$

Taking the first derivative with respect to a single parameter $\theta_k$ using the product rule yields:

$$\frac{\partial f(\boldsymbol{\theta})}{\partial \theta_k} = \frac{\partial \langle \psi(\boldsymbol{\theta})|}{\partial \theta_k} M \, |\psi(\boldsymbol{\theta})\rangle + \langle \psi(\boldsymbol{\theta})| \, M \frac{\partial \, |\psi(\boldsymbol{\theta})\rangle}{\partial \theta_k}$$

Since the observable $M$ is a Hermitian operator ($M = M^\dagger$), the two terms in the expression are complex conjugates of each other. Recall that for any complex number $z$, $z + z^* = 2\text{Re}(z)$. This allows for the simplification:

$$\frac{\partial f(\boldsymbol{\theta})}{\partial \theta_k} = 2\text{Re}\left( \langle \psi(\boldsymbol{\theta})| \, M \frac{\partial \, |\psi(\boldsymbol{\theta})\rangle}{\partial \theta_k} \right)$$

Using this first derivative term we can now find a term for the second derivative or hessian element $H_{kl}$.

**Lemma 3.** *The elements of the Hessian matrix, $H_{kl} = \frac{\partial^2 f(\boldsymbol{\theta})}{\partial \theta_l \partial \theta_k}$, are given by the expression:*

$$H_{kl} = 2Re\left( \frac{\partial \langle \psi(\boldsymbol{\theta})|}{\partial \theta_l} M \frac{\partial \, |\psi(\boldsymbol{\theta})\rangle}{\partial \theta_k} + \langle \psi(\boldsymbol{\theta})| \, M \frac{\partial^2 \, |\psi(\boldsymbol{\theta})\rangle}{\partial \theta_l \partial \theta_k} \right)$$

*Proof.* We begin with the simplified expression for the first partial derivative:

$$\frac{\partial f(\boldsymbol{\theta})}{\partial \theta_k} = 2\text{Re}\left( \langle \psi(\boldsymbol{\theta})| \, M \frac{\partial \, |\psi(\boldsymbol{\theta})\rangle}{\partial \theta_k} \right)$$

To find the Hessian element $H_{kl}$, we differentiate this expression with respect to another parameter, $\theta_l$:

$$H_{kl} = \frac{\partial}{\partial \theta_l} \left[ 2\text{Re}\left( \langle \psi(\boldsymbol{\theta})| \, M \frac{\partial \, |\psi(\boldsymbol{\theta})\rangle}{\partial \theta_k} \right) \right]$$

Since differentiation and the real part operator are both linear operations, their order can be interchanged:

$$H_{kl} = 2\text{Re}\left[ \frac{\partial}{\partial \theta_l} \left( \langle \psi(\boldsymbol{\theta})| \, M \frac{\partial \, |\psi(\boldsymbol{\theta})\rangle}{\partial \theta_k} \right) \right]$$

We now apply the product rule to the term inside the brackets:

$$\frac{\partial}{\partial\theta_l}\left(\langle\psi(\boldsymbol{\theta})|\,M\frac{\partial\,|\psi(\boldsymbol{\theta})\rangle}{\partial\theta_k}\right)=\left(\frac{\partial\,\langle\psi(\boldsymbol{\theta})|}{\partial\theta_l}\right)\left(M\frac{\partial\,|\psi(\boldsymbol{\theta})\rangle}{\partial\theta_k}\right)+\langle\psi(\boldsymbol{\theta})|\,M\left(\frac{\partial^2\,|\psi(\boldsymbol{\theta})\rangle}{\partial\theta_l\partial\theta_k}\right)$$

Substituting this result back into the expression for $H_{kl}$ yields the final form, which matches the expression used in the main text:

$$H_{kl}=2\mathrm{Re}\left(\frac{\partial\,\langle\psi(\boldsymbol{\theta})|}{\partial\theta_l}M\frac{\partial\,|\psi(\boldsymbol{\theta})\rangle}{\partial\theta_k}+\langle\psi(\boldsymbol{\theta})|\,M\frac{\partial^2\,|\psi(\boldsymbol{\theta})\rangle}{\partial\theta_l\partial\theta_k}\right)$$

This completes the derivation. $\qquad\square$

### B.2 HESSIAN VIA THE PARAMETER-SHIFT RULE – PRACTICAL COMPUTE ON QUANTUM HARDWARE

An alternative to direct analytical differentiation for finding the Hessian elements is to apply the parameter-shift rule successively. This method is particularly relevant for executing on quantum hardware, as it expresses the second derivatives in terms of expectation values of the original circuit with shifted parameters. The parameter-shift rule is applicable for gates of the form $U_k(\theta_k)=e^{-i\theta_k G_k}$, where the generator $G_k$ has two unique eigenvalues, $\pm r_k$ Schuld et al. (2019); Wierichs et al. (2022).

The first derivative of the objective function $f(\boldsymbol{\theta})$ with respect to a parameter $\theta_k$ is given by:

$$\frac{\partial f(\boldsymbol{\theta})}{\partial\theta_k}=r_k\left[f(\boldsymbol{\theta}_k^+)-f(\boldsymbol{\theta}_k^-)\right]$$

where $f(\boldsymbol{\theta}_k^{\pm})$ denotes an evaluation of the function with the $k$-th parameter shifted by an amount $s_k=\frac{\pi}{4r_k}$, i.e., $\theta_k\to\theta_k\pm s_k$. For standard single-qubit Pauli rotation gates ($R_x,R_y,R_z$), the generator is $G_k=\frac{1}{2}P_i$, which has eigenvalues $\pm\frac{1}{2}$. This results in $r_k=1/2$ and a shift of $s_k=\pi/2$. To find the Hessian elements $H_{kl}=\frac{\partial^2 f(\boldsymbol{\theta})}{\partial\theta_l\partial\theta_k}$, we differentiate the expression for the first derivative. We consider two cases.

#### B.2.1 CASE 1: OFF-DIAGONAL ELEMENTS ($k\neq l$)

We differentiate the first derivative with respect to a different parameter $\theta_l$:

$$H_{kl}=\frac{\partial}{\partial\theta_l}\left(\frac{\partial f(\boldsymbol{\theta})}{\partial\theta_k}\right)=\frac{\partial}{\partial\theta_l}\left[r_k\left(f(\boldsymbol{\theta}_k^+)-f(\boldsymbol{\theta}_k^-)\right)\right]$$

Since differentiation is a linear operator, we can bring it inside the brackets:

$$H_{kl}=r_k\left[\frac{\partial f(\boldsymbol{\theta}_k^+)}{\partial\theta_l}-\frac{\partial f(\boldsymbol{\theta}_k^-)}{\partial\theta_l}\right]$$

We now apply the parameter-shift rule again for the derivative with respect to $\theta_l$ on each term. The shift for $\theta_l$ is $s_l=\frac{\pi}{4r_l}$.

$$\frac{\partial f(\boldsymbol{\theta}_k^+)}{\partial\theta_l}=r_l\left[f(\ldots,\theta_k+s_k,\ldots,\theta_l+s_l,\ldots)-f(\ldots,\theta_k+s_k,\ldots,\theta_l-s_l,\ldots)\right]$$

$$\frac{\partial f(\boldsymbol{\theta}_k^-)}{\partial\theta_l}=r_l\left[f(\ldots,\theta_k-s_k,\ldots,\theta_l+s_l,\ldots)-f(\ldots,\theta_k-s_k,\ldots,\theta_l-s_l,\ldots)\right]$$

Substituting these back and combining terms, we arrive at the general expression for the off-diagonal elements of the Hessian:

$$\boxed{H_{kl}=r_k r_l\left[f(\boldsymbol{\theta}_{k,l}^{++})-f(\boldsymbol{\theta}_{k,l}^{+-})-f(\boldsymbol{\theta}_{k,l}^{-+})+f(\boldsymbol{\theta}_{k,l}^{--})\right]}\tag{13}$$

where $f(\boldsymbol{\theta}_{k,l}^{\alpha\beta})$ denotes evaluating $f$ with $\theta_k$ shifted by $\alpha s_k$ and $\theta_l$ shifted by $\beta s_l$, with $\alpha,\beta\in\{+,-\}$. For standard Pauli rotations, where $r_k=r_l=1/2$ and $s_k=s_l=\pi/2$, this becomes:

$$H_{kl}=\frac{1}{4}\left[f(\theta_k+\frac{\pi}{2},\theta_l+\frac{\pi}{2})-f(\theta_k+\frac{\pi}{2},\theta_l-\frac{\pi}{2})-f(\theta_k-\frac{\pi}{2},\theta_l+\frac{\pi}{2})+f(\theta_k-\frac{\pi}{2},\theta_l-\frac{\pi}{2})\right]$$

### B.2.2 Case 2: Diagonal Elements ($k = l$)

For the diagonal elements, we differentiate with respect to the same parameter $\theta_k$ twice. This is equivalent to applying a central finite difference formula to the first derivative:

$$H_{kk} = \frac{\partial^2 f(\boldsymbol{\theta})}{\partial \theta_k^2} = r_k \left[ \frac{\partial f(\boldsymbol{\theta}_k^+)}{\partial \theta_k} - \frac{\partial f(\boldsymbol{\theta}_k^-)}{\partial \theta_k} \right]$$

Applying the chain rule, $\frac{\partial f(\theta_k \pm s_k)}{\partial \theta_k}$ is simply the first derivative of $f$ evaluated at the shifted point. Using the parameter-shift rule for these terms gives:

$$\frac{\partial f(\boldsymbol{\theta}_k^+)}{\partial \theta_k} = r_k \left[ f(\ldots, (\theta_k + s_k) + s_k, \ldots) - f(\ldots, (\theta_k + s_k) - s_k, \ldots) \right] = r_k \left[ f(\boldsymbol{\theta}_k^{2+}) - f(\boldsymbol{\theta}) \right]$$

$$\frac{\partial f(\boldsymbol{\theta}_k^-)}{\partial \theta_k} = r_k \left[ f(\ldots, (\theta_k - s_k) + s_k, \ldots) - f(\ldots, (\theta_k - s_k) - s_k, \ldots) \right] = r_k \left[ f(\boldsymbol{\theta}) - f(\boldsymbol{\theta}_k^{2-}) \right]$$

where $\boldsymbol{\theta}_k^{2\pm}$ denotes a shift of $\theta_k \to \theta_k \pm 2s_k$. Substituting back, we obtain:

$$H_{kk} = r_k^2 \left[ f(\boldsymbol{\theta}_k^{2+}) - 2f(\boldsymbol{\theta}) + f(\boldsymbol{\theta}_k^{2-}) \right] \tag{14}$$

For standard Pauli rotations, $r_k = 1/2$ and $2s_k = \pi$, so the expression simplifies to:

$$H_{kk} = \frac{1}{4} \left[ f(\theta_k + \pi) - 2f(\theta_k) + f(\theta_k - \pi) \right]$$

### B.2.3 Compute Requirements – Using Quantum Hardware

These formulas show that every element of the Hessian can be calculated by evaluating the original objective function at different parameter values. This provides a practical, albeit potentially costly, method for calculating the Hessian, whose spectral norm is shown to be globally bounded by our main result. Practically speaking this means you would need a total of 4 forward passes to compute a single hessian element $H_{kl}$ at the shifts $\{s_k^+ s_l^+, s_k^+ s_l^-, s_k^- s_l^+, s_k^- s_l^-\}$. For the diagonal elements this reduces to 2 forward passes at the shifts $\{2s_k^+, 2s_l^-\}$. The total number of forward passes to compute a $P \times P$ hessian matrix for a parameter vector, $\boldsymbol{\theta}$, is then given by:

$$FP_{\boldsymbol{\theta}} = \mathcal{M}_s [4(P^2 - P) + 2P + 1] \tag{15}$$

Importantly on real quantum hardware each forward pass needs to be ran several times due to the quantum state collapse to get a final output distribution or objective value. The number of shots determines how many times you run the circuit where a higher value giving a more accurate measure of $f(\boldsymbol{\theta})$, this is denoted by $\mathcal{M}_s$ where the number of shots is typically set around: $\mathcal{M}_s \sim 1000$.

## C Empirical & Analytical Comparison with Prior Smoothness Bounds

### C.1 Tightness Analysis

The results from Experiment 4 confirm that our bound offers significant improvements over prior work in terms of tightness and applicability. As seen in Figure 2, for circuits with mixed-norm generators, our bound provides a far tighter estimate of the true curvature than the Liu et al. (2025) bound. For the 25% ratio of weighted and standard rotations our bound ratio $\tilde{L}_{max}/L_{upper}$ is 232.8% higher than the bound ratio from Liu et al. (2025). The Liu et al. (2025) bound scales with the single maximum generator norm, making it overly conservative when even one gate has a large norm. Our bound, by summing the individual contributions, more accurately reflects the landscape's geometry. We see similar trends when comparing our bound with the on from Gu et al. (2021) using the three different observables we see our ratio in the Ising model being 41.4% higher than the $\tilde{L}_{max}/L_{Gu}$ ratio.

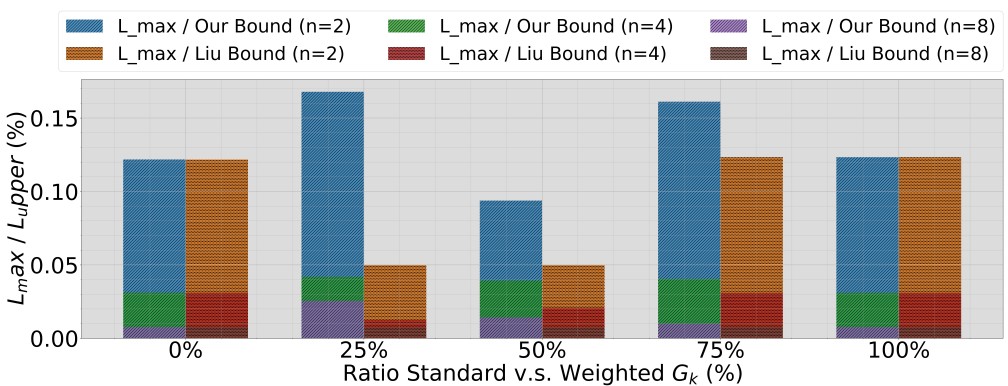

Figure 2: $\tilde{L}_{max}/L_{upper}$ ratio using our upper bound and Liu et al. (2025) bound with different ratios of weighted generators $G^W$ and standard Pauli gates.

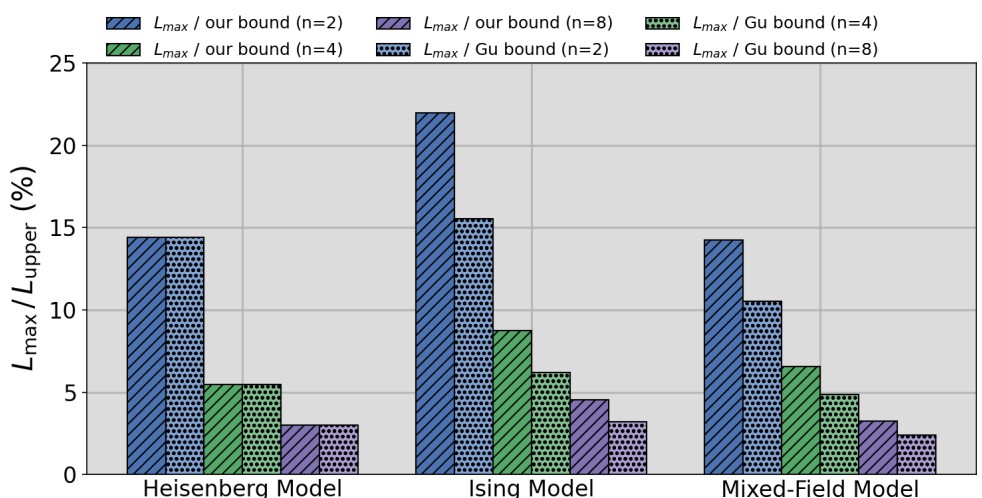

Figure 3: $\tilde{L}_{max}/L_{upper}$ ratio using our upper bound and Gu et al. (2021) bound with three different observables.

## C.2 Detailed Analytical and Empirical Comparison with Prior Smoothness Bounds

This section provides a formal mathematical comparison between our derived smoothness bound and the bounds from Gu et al. (2021) and Liu et al. (2025) seen in Table 1. We will prove that our bound is always at least as tight and quantify the **improvement factor** under various conditions, demonstrating its practical advantages, particularly for circuits with non-uniform gate generators or complex observables.

### C.2.1 Comparison with Gu et al. (2021)

The bound from Gu et al. (2021) is applicable to VQAs where parameterized gates are Pauli rotations. An observable $M$ can be decomposed into a weighted sum of $r$ Pauli strings $\rho_k$:

$$M = \sum_{k=1}^{r} \mu_k \rho_k$$

where $\mu_k$ are real coefficients.

- **Gu et al. (2021):** $L_{Gu} \leq P \sum_{k=1}^{r} |\mu_k|$

- **Our Bound (simplified for Pauli rotations):** As shown in Corollary 1, when all gates are standard Pauli rotations ($||G_k||_2 = 1/2$), our general bound simplifies to $L_{ours} \leq P||M||_2$.

**Proposition 1:** Our bound is always tighter than or equal to the Gu et al. (2021) bound.

*Proof.* We compare the core components of the bounds: $||M||_2$ and $\sum_k |\mu_k|$. By applying the triangle inequality to the operator norm and using the fact that the spectral norm of any Pauli string is one ($||\rho_k||_2 = 1$), we have:

$$||M||_2 = \left\| \sum_{k=1}^{r} \mu_k \rho_k \right\|_2 \leq \sum_{k=1}^{r} ||\mu_k \rho_k||_2 = \sum_{k=1}^{r} |\mu_k| \cdot ||\rho_k||_2 = \sum_{k=1}^{r} |\mu_k|$$

Since $||M||_2 \leq \sum_k |\mu_k|$, it follows directly that $P||M||_2 \leq P \sum_k |\mu_k|$, and thus $L_{ours} \leq L_{Gu}$. $\square$

To quantify the advantage, we define the **improvement factor** $\mathcal{I}_{Gu \to ours}$:

$$\mathcal{I}_{Gu \to ours} = \frac{L_{Gu}}{L_{ours}} = \frac{P \sum_{k=1}^{r} |\mu_k|}{P||M||_2} = \frac{\sum_{k=1}^{r} |\mu_k|}{||M||_2}$$

The tightness of our bound is determined by the gap in the triangle inequality. We can characterize the range of this improvement: $\mathcal{I}_{Gu \to ours} \in [1, r]$.

- **Case 1: Minimum Improvement ($\mathcal{I} = 1$)** The bounds are equal if $||M||_2 = \sum_k |\mu_k|$. This occurs in specific cases, such as when all Pauli strings in the decomposition of $M$ commute and share an eigenbasis (e.g., $M = \mu_1 Z_1 + \mu_2 Z_2$). In this scenario, the eigenvalues of $M$ are sums of the eigenvalues of the Pauli strings, and the largest eigenvalue's magnitude equals the sum of the coefficient magnitudes.

- **Case 2: Significant Improvement ($\mathcal{I} > 1$)** For many physically relevant Hamiltonians, the Pauli terms do not commute. Consider an observable composed of $r$ pairwise anti-commuting Pauli strings, each with equal weight $|\mu_k| = \mu$. In this case, $M^2 = (\mu \sum_{k=1}^{r} \rho_k)^2 = \mu^2 (rI) = r\mu^2 I$. The eigenvalues of $M$ are $\pm\mu\sqrt{r}$, so the spectral norm is $||M||_2 = \mu\sqrt{r}$. The improvement factor is:

$$\mathcal{I}_{Gu \to ours} = \frac{r \cdot \mu}{\mu\sqrt{r}} = \sqrt{r}$$

For problems like the transverse-field Ising model on $n$ qubits, where the Hamiltonian involves a sum of $\sim 2n$ non-commuting Pauli terms, our bound can be tighter by a factor of $\Theta(\sqrt{n})$.

### C.2.2 COMPARISON WITH LIU ET AL. (2025)

This comparison applies to general VQA circuits, not just those with Pauli rotations.

- **Liu et al. (2025):** $L_{Liu} \leq 4||M||_2 \cdot P \cdot (\max_k ||G_k||_2)^2$
- **Our Bound:** $L_{ours} \leq 4||M||_2 \sum_{k=1}^{P} ||G_k||_2^2$

**Proposition 2:** Our bound is always tighter than or equal to the Liu et al. (2025) bound.

*Proof.* Let $g_k = ||G_k||_2^2$. The sum of these values is $\sum_{k=1}^{P} g_k$. The maximum value is $\max_k g_k$. It is a basic inequality that for any set of non-negative numbers $\{g_k\}$:

$$\sum_{k=1}^{P} g_k \leq \sum_{k=1}^{P} (\max_j g_j) = P \cdot (\max_j g_j)$$

Substituting $g_k = ||G_k||_2^2$, we get $\sum_{k=1}^{P} ||G_k||_2^2 \leq P \cdot (\max_k ||G_k||_2)^2$. Multiplying by the common factor $4||M||_2$ preserves the inequality, proving $L_{ours} \leq L_{Liu}$. $\square$

The **improvement factor** is the ratio of the two bounds:

$$\mathcal{I}_{Liu \to ours} = \frac{L_{Liu}}{L_{ours}} = \frac{4||M||_2 \cdot P \cdot (\max_k ||G_k||_2)^2}{4||M||_2 \sum_{k=1}^{P} ||G_k||_2^2} = \frac{P \cdot (\max_k ||G_k||_2)^2}{\sum_{k=1}^{P} ||G_k||_2^2}$$

The improvement factor lies in the interval $\mathcal{I}_{Liu \to ours} \in [1, P]$.

- **Case 1: Minimum Improvement ($\mathcal{I} = 1$)** The bounds are identical if and only if all generator norms are equal, i.e., $||G_k||_2 = c$ for all $k$. This corresponds to a circuit with **uniform generators**. In this case, $\sum_k ||G_k||_2^2 = P \cdot c^2$ and $P \cdot (\max_k ||G_k||_2)^2 = P \cdot c^2$, making the ratio 1.

- **Case 2: Maximum Improvement ($\mathcal{I} \to P$)** The bound from Liu et al. becomes least tight for circuits with **heterogeneous generators**, where one or a few generators have a significantly larger norm than the others. Consider a circuit where one generator has a large norm-squared, $||G_1||_2^2 = C_{max}$, while the other $P - 1$ generators have a very small norm-squared, $||G_{k>1}||_2^2 = \epsilon \to 0$. The improvement factor becomes:

$$\mathcal{I}_{Liu \to ours} = \frac{P \cdot C_{max}}{C_{max} + (P - 1)\epsilon} \xrightarrow{\epsilon \to 0} \frac{P \cdot C_{max}}{C_{max}} = P$$

Our bound can be up to $P$ times tighter in such highly non-uniform scenarios.

# D    SPECIFIC DETAILS: EXPERIMENTAL SETUP

## D.1    QUANTIFYING ANSATZ EXPRESSIBILITY

In Experiment 3, we investigate the connection between the geometric properties of the optimization landscape and the ansatz's "expressibility". Expressibility is a measure of a parameterized quantum circuit's (PQC) ability to generate quantum states that are well-representative of the entire Hilbert space Du et al. (2022); Sim et al. (2019). A highly expressive ansatz can explore a large volume of the state space, which is often considered a desirable trait for tackling a wide range of problems. To quantitatively measure this property, we adopt the statistical methodology proposed by Sim et al. (2019).

### D.1.1    MATHEMATICAL FRAMEWORK

The core idea is to quantify the difference between the distribution of states generated by a PQC and the uniform distribution of states over the Hilbert space, known as the ensemble of Haar-random states. This comparison is performed not on the states themselves, but on the distribution of fidelities between pairs of states. Let $|\psi(\boldsymbol{\theta})\rangle$ and $|\psi(\boldsymbol{\phi})\rangle$ be two quantum states generated by the same PQC but with independently sampled parameter vectors $\boldsymbol{\theta}$ and $\boldsymbol{\phi}$. The fidelity between these two pure states is given by:

$$F = |\langle \psi(\boldsymbol{\theta})|\psi(\boldsymbol{\phi})\rangle|^2 \tag{16}$$

For the ensemble of Haar-random states, the probability density function of these fidelities has a known analytical form:

$$P_{\text{Haar}}(F) = (N - 1)(1 - F)^{N-2} \tag{17}$$

where $N = 2^n$ is the dimension of the Hilbert space for an $n$-qubit system. The expressibility of a PQC, which we denote as $Expr$, is then defined as the Kullback-Leibler (KL) divergence between the PQC's empirically estimated fidelity distribution, $\hat{P}_{PQC}(F)$, and the theoretical Haar distribution, $P_{\text{Haar}}(F)$:

$$Expr = D_{KL}(\hat{P}_{PQC}(F; \boldsymbol{\theta})||P_{\text{Haar}}(F)) = \int_0^1 \hat{P}_{PQC}(F; \boldsymbol{\theta}) \log\left(\frac{\hat{P}_{PQC}(F; \boldsymbol{\theta})}{P_{\text{Haar}}(F)}\right) dF \tag{18}$$

A lower $D_{KL}$ value signifies higher expressibility, as it indicates that the distribution of states generated by the PQC is statistically closer to the uniform Haar distribution.

### D.1.2    NUMERICAL ESTIMATION

In practice, the continuous distribution $\hat{P}_{PQC}(F)$ must be estimated from a finite number of samples. Our implementation follows the procedure outlined by Sim et al. (2019) and is realized in our code by the `calculate_expressibility_kld` function. The process involves four main steps:

1. **Parameter Sampling**: We generate $S$ pairs of parameter vectors, $\{(\boldsymbol{\theta}_i, \boldsymbol{\phi}_i)\}_{i=1}^{S}$. Each element of every vector is drawn independently from a uniform distribution over $[0, 2\pi]$. In our experiments, we used $S = 2000$ sample pairs.

2. **State Generation and Fidelity Calculation**: For each parameter vector pair $(\boldsymbol{\theta}_i, \boldsymbol{\phi}_i)$, we execute the ansatz to produce the corresponding state vectors $|\psi(\boldsymbol{\theta}_i)\rangle$ and $|\psi(\boldsymbol{\phi}_i)\rangle$. The fidelity $F_i = |\langle\psi(\boldsymbol{\theta}_i)|\psi(\boldsymbol{\phi}_i)\rangle|^2$ is then computed.

3. **Distribution Discretization**: To approximate the integral, we discretize the fidelity domain $[0, 1]$ into $k = 75$ bins, matching the methodology in the reference study. We then construct a normalized histogram of the $S$ calculated fidelities to produce a discrete probability distribution for the PQC, $\hat{P}_{PQC}$. The theoretical Haar distribution, $P_{\text{Haar}}$, is also evaluated at the center of these bins to create a corresponding discrete distribution.

4. **KL Divergence Computation**: With the two discrete probability distributions, the KL divergence is calculated as a sum:

$$D_{KL} = \sum_{j=1}^{k} \hat{P}_{PQC}(\text{bin}_j) \log\left(\frac{\hat{P}_{PQC}(\text{bin}_j)}{P_{\text{Haar}}(\text{bin}_j)}\right) \tag{19}$$

where $j$ indexes the histogram bins.

This numerical procedure provides the $D_{KL}$ value used in our analysis to track how the expressibility of our ansatz changes with circuit depth.

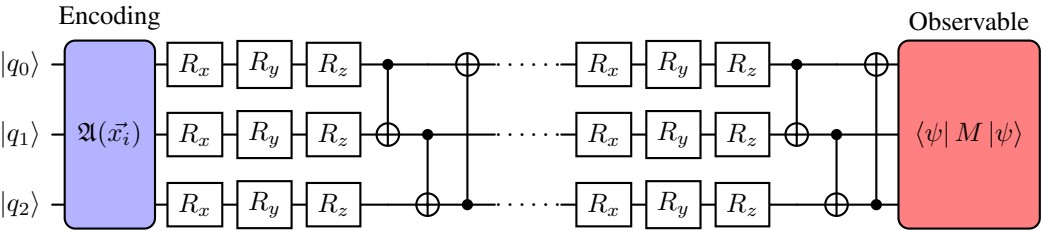

Figure 4: Standard VQA architecture with data encoding, variational layers, and measurement. The circuit begins with an encoding block $\mathfrak{A}(\vec{x_i})$ that maps classical data to quantum states, followed by alternating parametrized rotations ($R_x$, $R_y$, $R_z$) and entangling CNOT gates arranged in layers. The final measurement observable $\langle\psi| M |\psi\rangle$ extracts the computational result. This circuit represents the one used in Section 5.

### D.2 Justification of Sampling Methodology for Curvature Estimation

To validate our theoretical bound, we conducted numerical experiments to isolate the impact of circuit depth/width ($P/n$), observable norm ($\|M\|_2$), and generator norms ($\|G_k\|_2$). A core component of this validation is estimating the true maximum landscape curvature, $\tilde{L}_{max}$.

Formally, the smoothness constant $L$ is determined by the global maximum of the Hessian's spectral norm over the entire parameter space: $\tilde{L}_{max} = \max_{\boldsymbol{\theta}\in\mathbb{R}^P} \|\nabla^2 f(\boldsymbol{\theta})\|_2$. The objective function is a mapping from a $P$-dimensional real vector space to a scalar, $f : \mathbb{R}^P \to \mathbb{R}$. As the number of parameters $P$ increases, the dimensionality of this landscape grows, and its complexity can increase exponentially. The number of local minima, maxima, and saddle points can become vast, making an analytical or numerical search for the single global maximum curvature infeasible. Therefore, we adopt a practical method: we estimate $\tilde{L}_{max}$ by sampling a large number of random parameter vectors and reporting the largest observed Hessian norm. This method provides a representative value for the maximum curvature that one would likely encounter during training. Our choice of 1000 random parameter samples, drawn uniformly from $\boldsymbol{\theta}_i \in [0, 2\pi]$, is justified by the following analyses:

**1. Convergence of the Estimate** To ensure our sample size is sufficient, we first conducted a convergence study. As shown in Figure 5a, we repeatedly estimated $\tilde{L}_{max}$ for circuits of varying widths while increasing the number of parameter samples from 10 to 4000. The plot clearly shows that

the estimated maximum curvature stabilizes well before 1000 samples. Our choice of 1000 thus places us firmly in the converged regime, ensuring a reliable estimate while maintaining computational tractability.

**2. Stability of the Estimate**    To confirm that our 1000-sample estimate is not an arbitrary result but a stable characteristic, we performed 10 independent runs for a 4-qubit VQA at three different depths: 1 layer (P=12), 5 layers (P=60), and 10 layers (P=120). The results are shown in Figure 7b, as seen:

- P=12 ($L_{upper} = 12$): The mean $\tilde{L}_{max}$ across 10 runs was **0.4997** with a standard deviation of only **0.00022**.

- P=60 ($L_{upper} = 60$): The mean $\tilde{L}_{max}$ was **1.851** with a standard deviation of **0.025**.

- P=120 ($L_{upper} = 120$): The mean $\tilde{L}_{max}$ was **3.210** with a standard deviation of **0.050**.

Despite the absolute standard deviation growing slightly with circuit depth (expected as growing the parameter space grows the potential curvature), it remains exceptionally small relative to the mean in all cases. This low variance demonstrates that our sampling methodology is robust and consistently captures the characteristic maximum curvature of a given ansatz architecture that could be expected to observe in practicality.

**3. Capturing Landscape Trends**    Finally, our sampling method effectively reveals key geometric trends. Figure 5b shows the full distribution of 1000 sampled Hessian norms for circuits with $n$=1, 2, and 4 qubits. As the number of qubits increases, the mean, median, and overall variance of the Hessian norms systematically decrease. The distributions become significantly more compressed around a much lower average value. This provides a clear statistical picture of the landscape flattening for wider circuits—a second-order signature of the barren plateau phenomenon—validating that our sampling is sufficient to capture these essential structural properties.

In summary, our methodology of using 1000 random samples provides a computationally feasible, convergent, and statistically stable approach to estimating the maximum landscape curvature.

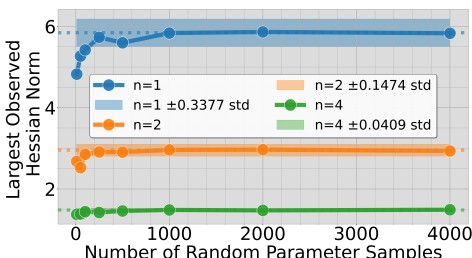 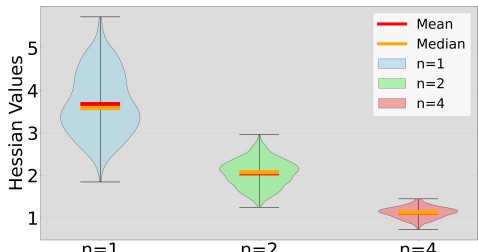

(a) Convergence of the estimated maximum Hessian norm ($\tilde{L}_{max}$) with an increasing number of parameter samples (10-4000) for circuits of varying width ($n = 1, 2, 4$).

(b) Distribution of sampled Hessian norms for circuits of varying widths ($n = 1, 2, 4$). The mean and variance of the curvature distribution decrease as the number of qubits increases.

### D.3    TESTING VARIOUS VQA ARCHITECTURE

A central finding of this work is that the maximum landscape curvature, $\tilde{L}_{max}$, grows in a predictable, linear fashion that our theoretical bound captures. As demonstrated in Section 5.2, this predictable scaling—observed as a stabilization of the $\tilde{L}_{max}/L_{upper}$ ratio—persists across variations in circuit depth ($P$), width ($n$), observable norms ($\|M\|_2$), and gate generator norms ($\|G_k\|_2$). To further substantiate the generality of this phenomenon, we extend our analysis to include additional VQA architectures that are structurally distinct and commonly employed in the literature.

We compare the baseline **Standard Ring** architecture from Section 5 (Figure 4) with two new designs that feature fundamentally different entanglement strategies:

1. **Linear CZ:** This architecture employs a linear entanglement topology, where qubit $i$ acts on qubit $i + 1$. Crucially, it uses a Controlled-Z (CZ) gate, which, unlike the CNOT gate,

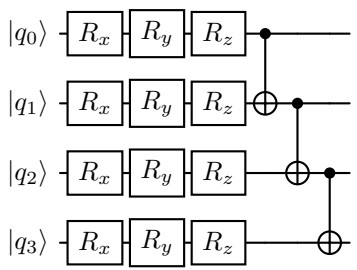 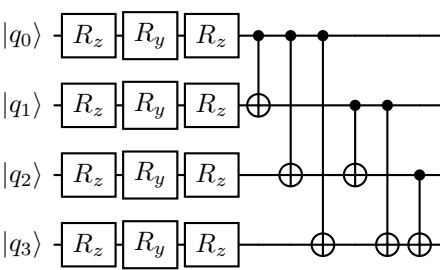

(a) Linearly entangled VQA for 4 qubits, with $CZ$ entanglement and $R_x, R_y, Rz$ rotations.

(b) Maximally entangled VQA for 4 qubits, with $CNOT$ entanglement and $R_z, R_y, R_z$ rotations.

Figure 6: Additional VQA architectures studied for Experiment 1. We observe the same plateau behavior of $\tilde{L}_{max}/L_{upper}$ in all cases.

> creates symmetric correlations between qubits. This design is often used for its hardware efficiency on certain quantum computing platforms. An example is shown in Figure 6a.

2. **All-to-All CNOT:** This architecture implements a maximal entanglement strategy, connecting every qubit to every other qubit within a layer using CNOT gates. While computationally expensive, this topology facilitates the rapid scrambling of quantum information, allowing for the creation of highly complex correlations across the entire system. We also vary the parameterized rotations in this model to use an $R_z, R_y, R_z$ sequence, as illustrated in Figure 6b.

For this analysis, we repeated the core experiment from the main text: for each architecture, we fixed the number of qubits to $n = 2$ and $n = 4$ and progressively increased the circuit depth, extending up to a total of $P = 120$ parameters. Our findings, presented in Figure 7a, confirm that the stabilization of the $\tilde{L}_{max}/L_{upper}$ ratio is a robust and architecture-independent phenomenon. While the initial behavior and the exact value of the stabilized ratio differ between architectures—with the Linear CZ circuit showing an initial increase in relative curvature while the others show a decrease—all designs converge to a stable, constant ratio at sufficient depth. This result strongly reinforces our central claim: the "plateau effect" is not an artifact of a specific circuit design but rather a fundamental geometric property of VQA optimization landscapes. We theorize this behavior is intrinsically linked to expressibility saturation and parameter redundancy, phenomena that are universal to such models. The predictable scaling of landscape curvature across these diverse architectures validates the broad applicability of the practical tools derived from it, namely the learning rate calibration heuristic and the use of the curvature ratio as a diagnostic for inefficient overparameterization.

### D.4 SOFTWARE AND TOOLS

All numerical simulations were implemented in Python (version 3.9.20), utilizing the PennyLane library Bergholm et al. (2018) (version 0.38.0) to construct, simulate, and differentiate the quantum circuits. The Hessian matrix for each parameter sample was computed using PennyLane's automatic differentiation capabilities, specifically by composing the gradient and Jacobian functions (`qml.jacobian(qml.grad(...))`). Data visualization was performed using Matplotlib and Seaborn. The experiments were executed on a high-performance computing server equipped with dual Intel(R) Xeon(R) Gold 6258R CPUs, 1.5 TB of RAM, and eight NVIDIA A100 (40GB) GPUs. The simulations were run exclusively on the CPU via PennyLane's `default.qubit` device. This was a necessary choice, as calculating the second-order derivatives required for our analysis is not supported on PennyLane's GPU-accelerated devices `lightning.gpu`. All source code is made available in the supplementary materials with public release available on GitHub upon acceptance.

### E WHEN L-SMOOTHNESS BREAKS

The L-smoothness property, which we prove holds for standard Variational Quantum Algorithms (VQAs), is not guaranteed if the architecture includes operations that introduce discontinuities with

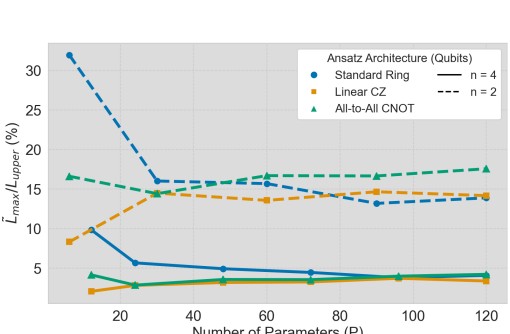 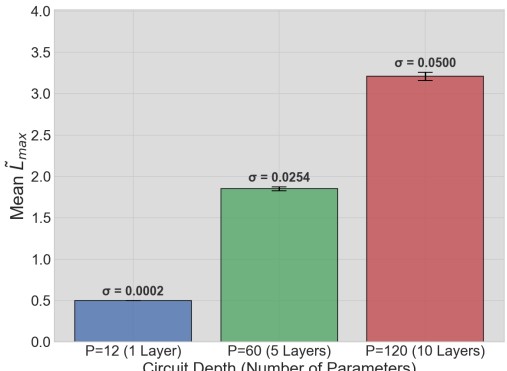

(a) Generality of Curvature Stabilization Across VQA Architectures. The ratio of $\tilde{L}_{max}/L_{upper}$, is plotted against the number of parameters (P) for three distinct ansatz designs. While the initial scaling and stabilized values differ, all architectures exhibit a plateau pattern, confirming that this geometric stabilization is a general phenomenon.

(b) Stability of Maximum Curvature Estimation. The plot shows the mean estimated maximum curvature ($\tilde{L}_{max}$) for a 4-qubit circuit at three different depths, averaged over 10 independent runs of 1000 parameter samples each. The error bars, representing $\pm 1$ standard deviation, demonstrate the high consistency and robustness of our sampling methodology.

respect to the circuit parameters. A canonical example would be a parameterized gate whose rotation angle is determined by a non-smooth function of a variational parameter $\theta_k$. Formally, consider a gate $U_k$ where the angle is governed by a function $g(\theta_k)$ analogous to the Rectified Linear Unit (ReLU) activation function common in classical neural networks Nair & Hinton (2010):

$$U_k(g(\theta_k)) \quad \text{where} \quad g(\theta_k) = \max(0, \theta_k - \tau) \tag{20}$$

or equivalently, using the Heaviside step function $\Theta(\cdot)$:

$$g(\theta_k) = (\theta_k - \tau) \cdot \Theta(\theta_k - \tau) \tag{21}$$

This conditional logic makes the objective function $f(\theta)$ non-differentiable at the threshold point $\theta_k = \tau$, violating the conditions for L-smoothness. It is important to note that such constructs are highly uncommon in the VQA literature. VQAs achieve non-linearity through the sinusoidal nature of quantum gate operations, obviating the need for explicit non-linear activation functions applied to the parameters themselves. A notable architecture that might initially appear to violate L-smoothness is the Quantum Convolutional Neural Network (QCNN) Cong et al. (2019). The QCNN pooling layer introduces a form of conditional logic: a fraction of qubits are measured, and the classical outcomes of these measurements determine which unitary rotations ($V_j$) are applied to adjacent qubits. This process could be misinterpreted as creating a discontinuity. However, the QCNN architecture remains L-smooth with respect to its variational parameters. The key distinction is that the conditional logic does not depend on a *variational parameter* $\theta_k$ crossing a threshold. Instead, it depends on the outcome of a quantum measurement, which is a stochastic event. The objective function $f(\theta)$ is an expectation value, which averages over all possible measurement outcomes. Let's consider a single pooling operation. Suppose a qubit is measured, yielding outcome $m \in \{0, 1\}$, and a conditional unitary $V_m$ is applied. The unitaries $V_0$ and $V_1$ are themselves parameterized by smooth parameters. The objective function, which is an expectation value, is a sum over the probabilities of these outcomes, weighted by the resulting state expectation:

$$f(\theta) = P(m=0|\theta) \cdot \langle\psi_{m=0}(\theta)|M|\psi_{m=0}(\theta)\rangle + P(m=1|\theta) \cdot \langle\psi_{m=1}(\theta)|M|\psi_{m=1}(\theta)\rangle \tag{22}$$

Here, $|\psi_m(\theta)\rangle$ is the state after the conditional unitary $V_m$ has been applied. Both the probabilities $P(m|\theta)$ and the final state vectors $|\psi_m(\theta)\rangle$ are smooth functions of the parameters $\theta$ that define the preceding circuit layers and the conditional unitaries $V_m$. Since $f(\theta)$ is a sum of products of smooth functions, it remains a smooth, twice-differentiable function of $\theta$. This logic aligns with our inductive proof in Lemma 1; because the operations before and after the measurement are standard parameterized gates, the overall objective function remains a multivariate trigonometric polynomial. Therefore, the measurement-based conditional logic of QCNNs does not break the L-smoothness property. To the best of our knowledge, VQA architectures employing parameter-dependent, non-differentiable functions like the ReLU example have not yet been proposed in the literature.

# F SETTING OPTIMAL LEARNING RATES

To estimate the scaling factor, one can perform a one-time pre-computation:

1. Sample a set of $S$ random parameter vectors, $\{\boldsymbol{\theta}_1, \boldsymbol{\theta}_2, ..., \boldsymbol{\theta}_S\}$.

2. For each sample $\boldsymbol{\theta}_i$, compute the Hessian matrix $\nabla^2 f(\boldsymbol{\theta}_i)$ and its spectral norm $||\nabla^2 f(\boldsymbol{\theta}_i)||_2$.

3. Approximate $\tilde{L}_{max}$ with the largest observed norm: $\tilde{L}_{max} \approx \max_i\{||\nabla^2 f(\boldsymbol{\theta}_i)||_2\}$.

4. The estimated scaling factor is then $\kappa_{\text{est}}(n) = (\max_i ||\nabla^2 f(\boldsymbol{\theta}_i)||_2)/L_{upper}$.

## F.1 DEMONSTRATED PREDICTIVE POWER AND COMPUTATIONAL SAVINGS

To validate the practical utility of using the $\kappa(n)$ scaling factor as a predictive tool for estimating curvature in deep circuits, we conducted a numerical experiment for 1, 2, and 4-qubit circuits. The methodology was as follows:

1. For each circuit width $n$, we first established a stable $\kappa(n)$ value by computing it for a single, moderately shallow "calibration circuit" ($D_{\text{cal}} = 10$ layers for $n$=1, and $D_{\text{cal}} = 5$ layers for $n$=2, 4). This step requires a one-time, upfront computation of $\tilde{L}_{\text{max}}$ for this smaller circuit.

2. We then used this single, fixed $\kappa(n)$ value to predict the effective smoothness constant ($L_{\text{effective}} = \kappa(n) \cdot L_{\text{upper}}$) and the corresponding optimal learning rate ($\eta^*_{\text{global}} \approx 1/L_{\text{effective}}$) for a range of much deeper target circuits.

Figure 8 demonstrates the heuristic's accuracy. The predicted learning rate (dashed line) closely tracks the ideal learning rate (solid line) across all tested qubit widths and circuit depths. The average prediction error remains remarkably low, consistently falling between **6-9%**. This confirms that our calibration method is not a reliable and data-driven strategy for setting a near-optimal learning rate for deep circuits, sidestepping the need for costly hyperparameter sweeps. The primary advantage of this heuristic is the immense reduction in computational cost. The number of circuit executions required to estimate $\tilde{L}_{\text{max}}$ scales quadratically with the number of parameters, $P$. Using the parameter-shift rule, the cost is given by $FP_{\boldsymbol{\theta}} = \mathcal{M}_s \cdot S \cdot (4P^2 - 2P + 1)$, where $S$ is the number of random parameter samples and $\mathcal{M}_s$ is the number of shots per evaluation. For simplicity, we can say the cost is proportional to $P^2$. We can quantify the savings with concrete examples from our 1-qubit experiment, where we used a 10-layer circuit to predict the learning rates for much deeper circuits.

- **Target Circuit (D=40):** The circuit has $P_{\text{target}} = 40 \times 1 \times 3 = 120$ parameters. The computational cost is proportional to $P_{\text{target}}^2 = 120^2 = 14,400$.

- **Our Heuristic (Calibration Circuit, D=10):** The calibration circuit has $P_{\text{cal}} = 10 \times 1 \times 3 = 30$ parameters. The cost is proportional to $P_{\text{cal}}^2 = 30^2 = 900$.

By performing the expensive computation on the much smaller calibration circuit, our heuristic achieves a computational speedup of over **16×** ($14,400/900$) for this scenario while giving a learning rate extremely close to the true optimal one. The quadratic nature of the cost means these savings grow dramatically. In the following section we will explore how important an optimal learning rate is for deep VQAs.

## F.2 VQE TRAINING WITH CALIBRATED LEARNING RATES

This section provides the implementation details and full results for the VQE training experiments discussed in Section 5.3. These experiments validate the effectiveness of our proposed learning rate calibration heuristic. The Variational Quantum Eigensolver (VQE) is a quantum-classical algorithm designed for near-term quantum devices. Its goal is to find the ground state energy of a given Hamiltonian—the lowest possible energy eigenvalue of a quantum system—by leveraging the variational principle Peruzzo et al. (2014); Kandala et al. (2017). We conducted VQE experiments for 1, 2, and 4-qubit systems to find the ground state energy of a target Hamiltonian. The architectures are as follows:

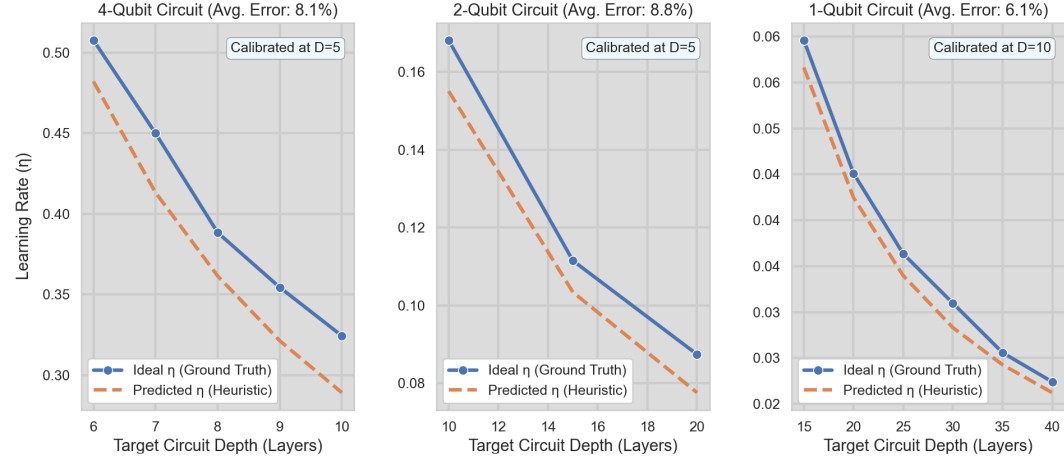

Figure 8: Empirical validation of the proposed learning rate heuristic: The predicted learning rate (dashed line), derived from a single calibration on a shallow circuit, accurately tracks the ideal learning rate (solid line, ground truth) for deeper target circuits across 1, 2, and 4-qubit systems. The low average prediction error (6-9%) demonstrates the heuristic's effectiveness for efficiently setting near-optimal learning rates.

1. 1-Qubit System: The ansatz consisted of $D = \{20, 30, 40\}$ layers of $(R_x, R_y, Rz)$ rotations. The Hamiltonian was $H = 0.6X_0 + 0.8Z_0$.

2. 2 and 4-Qubit Systems: The ansatz consisted of $D$ layers of $(R_x, R_y, Rz)$ rotations, followed by a chain of CNOT gates for entanglement. The Hamiltonian was the 1D transverse-field Ising model, $H = -\sum_i Z_i Z_{i+1} + 0.5 \sum_i X_i$. For the 2 qubit system $D = \{10, 15, 20\}$ and for the 4 qubit system $D = \{6, 8, 10\}$.

We calibrated the learning rates for the deep circuits by computing the scaling factor $\kappa(n)$ on a shallow circuit following. For the 1 qubit system we calibrated $\kappa(n)$ at $D = 10$ and for the 2 and 4 qubit system the scaling factor at $D = 5$. The optimal learning rate for the deep circuits was then set to $\eta^*_{global} = 1/(\kappa(n) \cdot L_{upper})$. For each target circuit depth we trained the VOEs using both the ADAM and SGD optimizers. We compared the performance of these optimizers using four different learning rates:

- Optimal $\eta$: the rate predicted by our calibration heuristic.

- High $\eta$: $5\times$ the optimal learning rate, chosen to test for instability.

- Low $\eta$: $0.2\times$ the optimal rate, chosen to test for slow convergence.

- Standard $\eta$: a commonly used default value like 0.01 for SGD and 0.001 for ADAM.

### F.2.1 VQE TRAINING RESULTS

The full training histories are shown in Figure 9 for SGD and Figure 10 for ADAM. A summary of the relative convergence times is presented in Table 2. Convergence is defined as finding the target ground state energy. The results clearly demonstrate the effectiveness of our calibration method. For the SGD optimizer the low and standard learning rates converges between 17%-608% slower compared to the optimal value based on VQE size. The high learning rate showed clear instability failing to converge completely within the training window. The oscillating training behavior of the high learning rate (where $\eta > 2/L$) is common due to the step size overshooting valleys. We see similar trends for the ADAM optimizer where the low learning rate converged 21% slower on average than the optimal value and the high value 54% slower than the optimal value. The standard learning rate values for ADAM in larger more complex systems ($n = 2, 4$) showed poor performance converging on average 1200% slower. These findings highlight the importance of selecting an appropriate learning rate, a task our heuristic makes more effective.

### F.3 FUTURE WORK: TOWARDS A GENERAL FORMULA FOR $\kappa(n)$

It is important to note that the proposed calibration heuristic is designed specifically for setting learning rates for **deep circuits**. The method's effectiveness relies on the observation that the scaling factor $\kappa(n)$ stabilizes after a certain number of layers. Consequently, if the target circuit is already very shallow (e.g., 1-5 layers), it may not be in the stabilized regime. In such cases, one cannot use an even shallower circuit to reliably estimate $\kappa(n)$, and the heuristic offers no advantage. The practitioner would be forced to either compute $\tilde{L}_{\max}$ directly for their circuit—the very task our method seeks to avoid—or resort to standard hyperparameter tuning. This limitation strongly motivates the future work of developing a general, analytical formula for $\kappa(n)$, which would provide a priori estimates without the need for any empirical calibration. Such a formula would provide *a priori* guidance on optimization without the need for numerical pre-computation. We hypothesize that $\kappa(n)$ is not merely a fitting parameter but is deeply connected to fundamental properties of the quantum circuit and the problem Hamiltonian. Our work strongly suggests that the behavior of $\kappa(n)$ is a second-order signature of the barren plateau phenomenon (McClean et al., 2018; Holmes et al., 2022). The observed exponential decay, $\kappa(n) \approx Ae^{-\gamma n}$, mirrors the exponential vanishing of gradient variance. As the Hilbert space grows, the landscape not only becomes flat on average (vanishing gradients) but its maximum curvature also collapses relative to its theoretical potential. A general theory for $\kappa(n)$ would therefore likely depend on the same factors known to cause barren plateaus:

- **Ansatz Expressibility:** Highly expressive circuits that approximate unitary 2-designs are more prone to barren plateaus (Holmes et al., 2022; Sim et al., 2019), which would correspond to a smaller $\kappa(n)$. The saturation of $\kappa(n)$ with depth (Figure 1c) directly reflects the saturation of expressibility.

- **Locality of the Observable:** Global observables are known to induce more severe barren plateaus than local ones (Cerezo et al., 2021). We thus expect the decay rate $\gamma$ to be significantly larger for global observables.

- **Entangling Structure:** The topology of entangling gates dictates the rate of information scrambling and thus the onset of the concentration effects underlying barren plateaus Ortiz Marrero et al. (2021); Patti et al. (2021).

A promising path towards a formula for $\kappa(n)$ involves applying tools from random matrix theory, specifically integration over the unitary group with respect to the Haar measure. While prior work has focused on the first and second moments of the gradient, one could aim to compute the expected value of the Hessian's spectral norm, $E_{U \sim \text{Haar}}[||\nabla^2 f(U)||_2]$. Though mathematically challenging, success in this endeavor would yield a foundational understanding of VQA landscape curvature and provide a powerful, predictive tool for algorithm design.

## USE OF LARGE LANGUAGE MODELS

In preparing this manuscript, we used large language models (Google Gemini Team et al. (2023)) as a writing aid for grammar correction, rephrasing, and enhancing readability. All scientific contributions, mathematical derivations, and experimental results are the original work of the authors, who assume full responsibility for the final content.

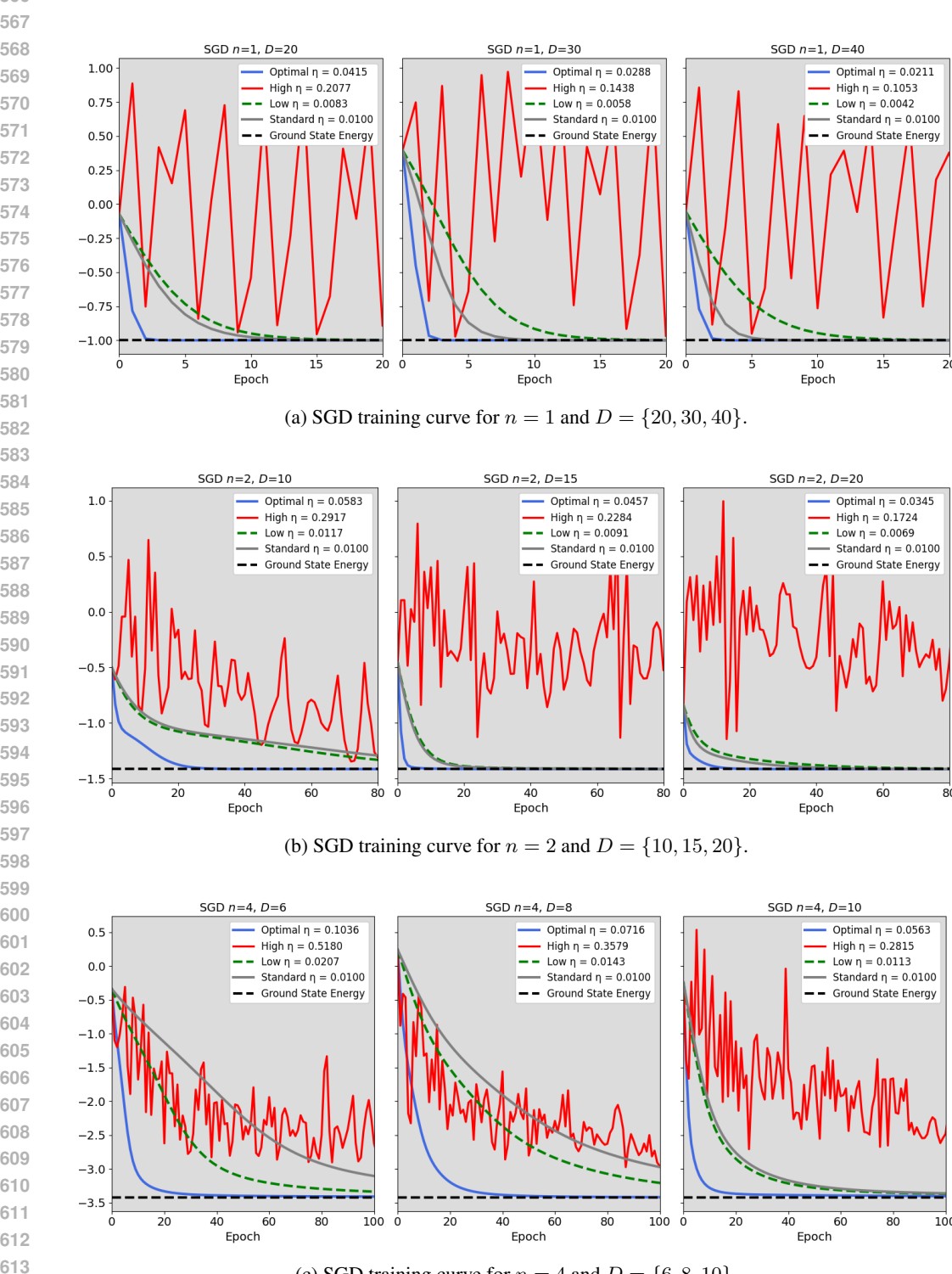

(a) SGD training curve for $n = 1$ and $D = \{20, 30, 40\}$.

(b) SGD training curve for $n = 2$ and $D = \{10, 15, 20\}$.

(c) SGD training curve for $n = 4$ and $D = \{6, 8, 10\}$.

Figure 9: Training curves for VQEs with $n = \{1, 2, 4\}$ qubits at different depths using the SGD optimizer with four different learning rates. The blue line represents the optimal learning rate calibrated using our heuristic. The black line shows the ground state energy of the Hamiltonian the VQE aims to reach.

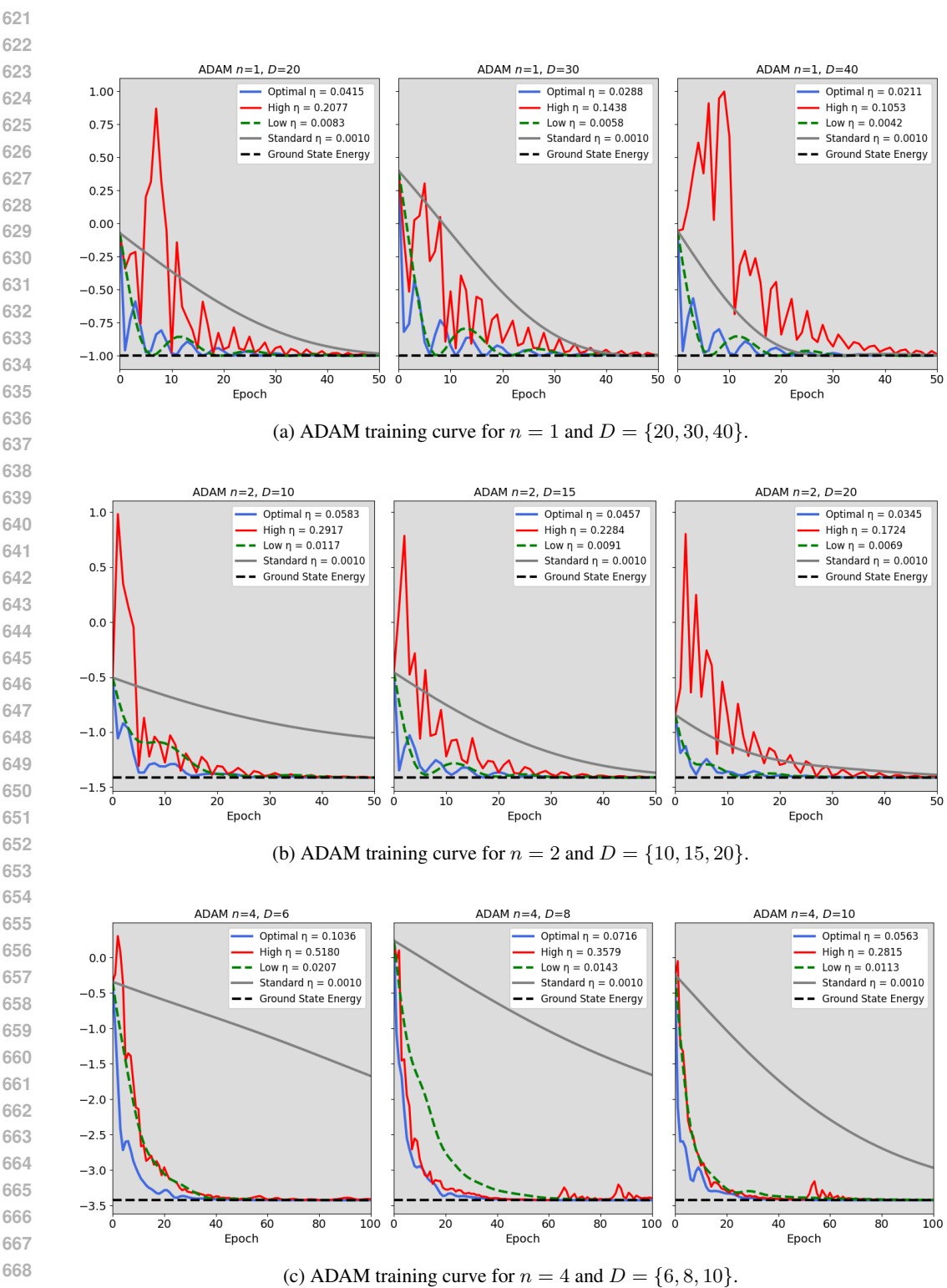

(a) ADAM training curve for $n = 1$ and $D = \{20, 30, 40\}$.

(b) ADAM training curve for $n = 2$ and $D = \{10, 15, 20\}$.

(c) ADAM training curve for $n = 4$ and $D = \{6, 8, 10\}$.

Figure 10: Training curves for VQEs with $n = \{1, 2, 4\}$ qubits at different depths using the ADAM optimizer with four different learning rates.

Figure 11: Training curves for noisy VQEs with $n = 4$ qubits at $D = \{6, 8, 10\}$ layers and comparison with QNG optimizer. All optimizer were run 5 times per learning rate and the dotted line shows the average convergence with the standard deviation.

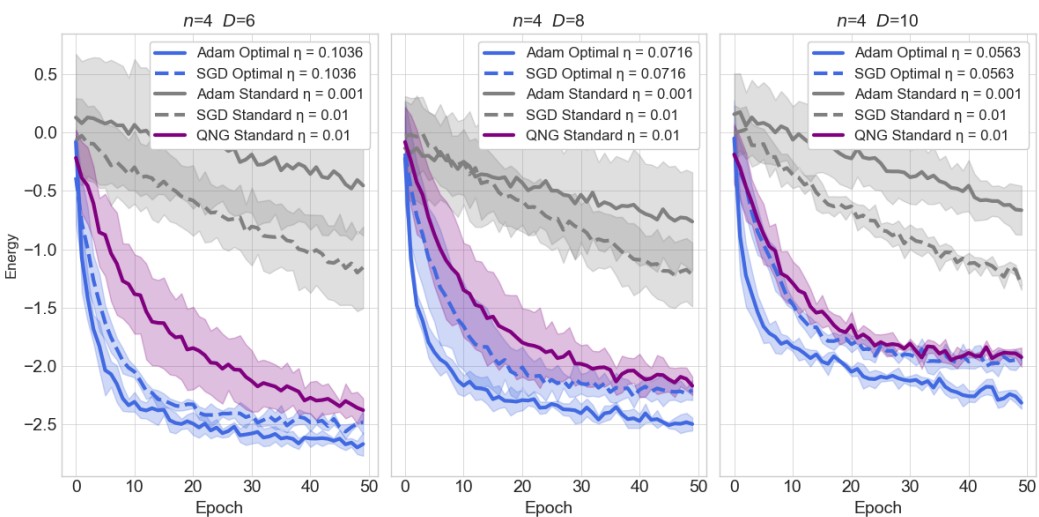

(a) Training curves using the QNG, ADAM, and SGD optimizers. Adam and SGD are run with their calibrated learning rates and standard learning rates. The calibrated learning rates using ADAM/SGD shows competitive performance with QNG while being computationally cheaper.

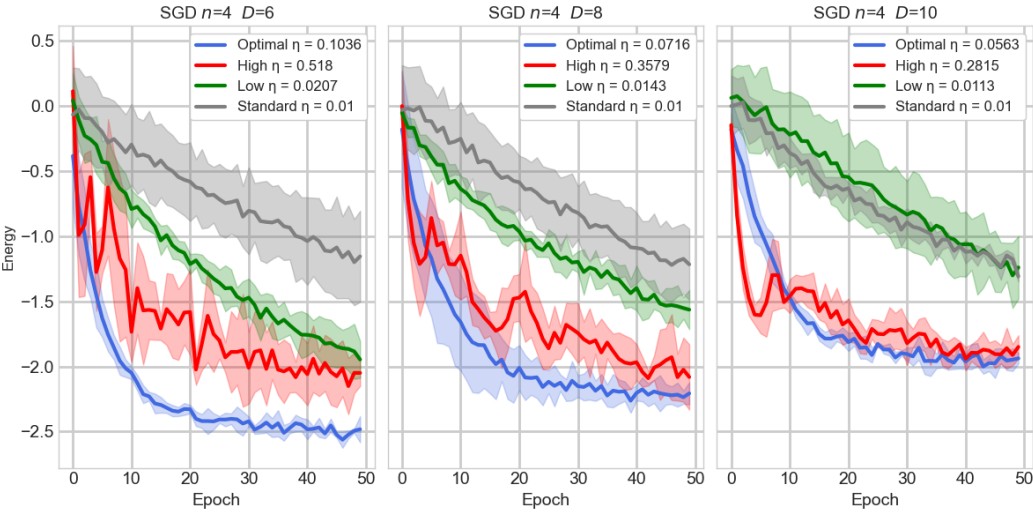

(b) Noisy SGD training curve for $n = 4$ and $D = \{6, 8, 10\}$. Five total runs per learning rate.

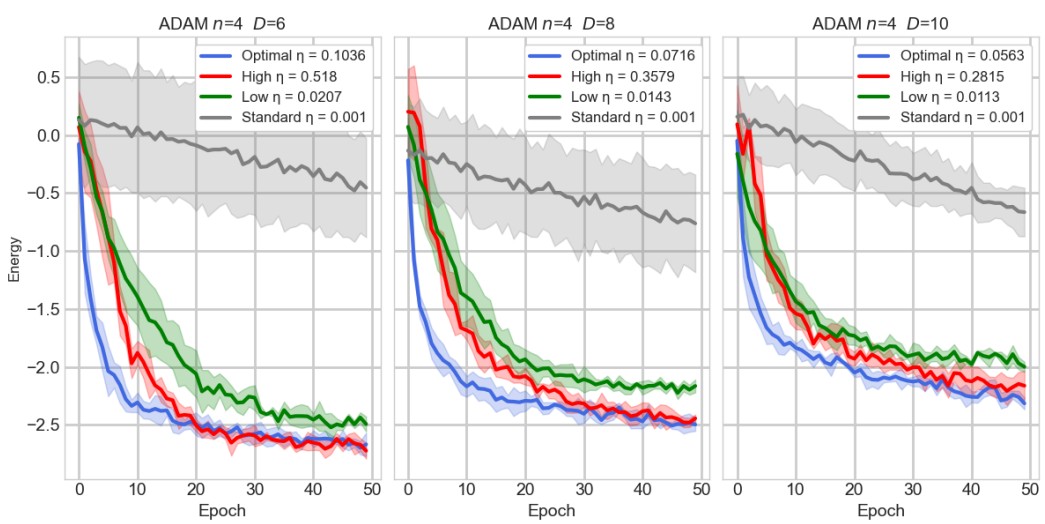

(c) Noisy ADAM training curve for $n = 4$ and $D = \{6, 8, 10\}$. Five total runs per learning rate.

Table 2: Relative convergence times (ratio of steps compared to optimal LR). $\infty$ indicates no convergence within tolerance for the 400 step training process for the given optimizer.

| System | Depth | High/Opt | Low/Opt | Std/Opt |
|--------|-------|----------|---------|---------|
| **SGD 1Q** | 20D | $\infty$ | $1.67\times$ | $1.58\times$ |
| | 30D | $\infty$ | $1.83\times$ | $1.42\times$ |
| | 40D | $\infty$ | $1.75\times$ | $1.17\times$ |
| **SGD 2Q** | 10D | $\infty$ | $3.39\times$ | $3.89\times$ |
| | 15D | $\infty$ | $1.85\times$ | $1.77\times$ |
| | 20D | $\infty$ | $2.25\times$ | $1.75\times$ |
| **SGD 4Q** | 6D | $\infty$ | $3.16\times$ | $6.08\times$ |
| | 8D | $\infty$ | $3.67\times$ | $5.12\times$ |
| | 10D | $\infty$ | $2.80\times$ | $3.10\times$ |
| **ADAM 1Q** | 20D | $1.47\times$ | $0.87\times$ | $1.77\times$ |
| | 30D | $1.55\times$ | $1.23\times$ | $1.52\times$ |
| | 40D | $1.73\times$ | $0.87\times$ | $1.07\times$ |
| **ADAM 2Q** | 10D | $1.46\times$ | $1.17\times$ | $7.08\times$ |
| | 15D | $1.48\times$ | $1.04\times$ | $2.24\times$ |
| | 20D | $2.39\times$ | $1.11\times$ | $2.61\times$ |
| **ADAM 4Q** | 6D | $1.48\times$ | $1.56\times$ | $15.84\times$ |
| | 8D | $1.11\times$ | $1.78\times$ | $13.56\times$ |
| | 10D | $1.22\times$ | $1.30\times$ | $6.70\times$ |

Table 3: Noisy VQE training data. Shows the energy reached for each optimizer with their specific learning rate after 10 and 50 epochs and their standard deviation during the five runs. ADAM with the calibrated learning rate reaches a lower ground state energy than QNG and it does so faster. Similarly the calibrated learning rate beats the other values like the standard ones.

| Optimizer | LR Setting | Depth | Energy Epoch 10 | Energy Epoch 50 | Mean STD |
|---|---|---|---|---|---|
| | Optimal $\eta$ | 6 | **-2.337** | -2.669 | 0.085 |
| | High $\eta$ | 6 | -1.970 | **-2.724** | 0.109 |
| | Low $\eta$ | 6 | -1.313 | -2.493 | 0.142 |
| | Standard (0.001) | 6 | -0.012 | -0.453 | 0.507 |
| | Optimal $\eta$ | 8 | **-2.124** | **-2.497** | 0.082 |
| Adam | High $\eta$ | 8 | -1.662 | -2.444 | 0.107 |
| | Low $\eta$ | 8 | -1.361 | -2.165 | 0.135 |
| | Standard (0.001) | 8 | -0.312 | -0.762 | 0.417 |
| | Optimal $\eta$ | 10 | **-1.790** | **-2.315** | 0.079 |
| | High $\eta$ | 10 | -1.494 | -2.163 | 0.113 |
| | Low $\eta$ | 10 | -1.347 | -1.999 | 0.117 |
| | Standard (0.001) | 10 | 0.032 | -0.665 | 0.246 |
| | Optimal $\eta$ | 6 | **-2.015** | **-2.482** | 0.087 |
| | High $\eta$ | 6 | -1.257 | -2.050 | 0.279 |
| | Low $\eta$ | 6 | -0.671 | -1.947 | 0.138 |
| | Standard (0.01) | 6 | -0.356 | -1.155 | 0.297 |
| | Optimal $\eta$ | 8 | **-1.600** | **-2.207** | 0.208 |
| SGD | High $\eta$ | 8 | -1.217 | -2.081 | 0.209 |
| | Low $\eta$ | 8 | -0.574 | -1.564 | 0.110 |
| | Standard (0.01) | 8 | -0.274 | -1.216 | 0.267 |
| | Optimal $\eta$ | 10 | -1.406 | **-1.938** | 0.089 |
| | High $\eta$ | 10 | **-1.507** | -1.850 | 0.122 |
| | Low $\eta$ | 10 | -0.179 | -1.240 | 0.277 |
| | Standard (0.01) | 10 | -0.303 | -1.309 | 0.098 |
| | Standard (0.01) | 6 | -1.298 | -2.378 | 0.259 |
| QNG | Standard (0.01) | 8 | -1.216 | -2.169 | 0.262 |
| | Standard (0.01) | 10 | -1.235 | -1.924 | 0.108 |

