# OpenReview forum: "VARIATIONAL QUANTUM ALGORITHMS ARE LIPSCHITZ SMOOTH"
_ICLR.cc/2026/Conference — Submitted to ICLR 2026_

### Official Review · Reviewer_ygvg · 2025-10-30

**Soundness:** 3
**Presentation:** 3
**Contribution:** 2
**Rating:** 6
**Confidence:** 5

**Summary:**

This paper presents a quite rigorous theoretical analysis of the L-smoothness property of VQA objective functions which scales linearly on the # of layers P, providing a worst-case upper limit on curvature that holds for general circuits. The authors provide a formal proof of global L-smoothness and derive an explicit upper bound on the smoothness constant L. Furthermore, they show that for certainc classes of VQAs this bound may take a very simple form, all the way down to be proportional to the depth of the underlying circuit. This result is then connected to circuits often assumed to be relevant for practical applications, including a diagnostic for ansatz overparameterization and a heuristic for setting near-optimal learning rates. While the contributions are welcome and well-supported, the analysis is confined to an idealized, noiseless setting, which limits the direct applicability of its conclusions to contemporary NISQ hardware.

This paper maybe useful since it can help better establish, for example, learning rates. $L$, provides an upper bound on the curvature and guarantees that the landscape is not infinitely "spiky and being able to guarantee this it is crucial for gradient-based methods because it ensures stability since if I know the maximum curvature, I can choose a learning rate small enough ($\eta \approx 1/L$) to guarantee that the optimization steps will not wildly overshoot a minimum.

However, to my view, this paper does not solve any of the ever present issues of VQAs. While this paper provides a valuable formalization of L-smoothness with the potential L-informed learning rate similar to many classical ML problems, the more fundamental and unresolved problem for VQAs is the lack of a meaningful lower bound on curvature, not the upper one, a condition that manifests as  the barren plateau problem where vanishing gradients render optimization intractable regardless of the landscape's theoretical smoothness.

**Strengths:**

(1) The paper formalizes a foundational property for VQAs. It is true that the VQA literature frequently relies on an implicit assumption of landscape smoothness to justify the use of gradient-based optimizers. This is well supported since by construction these are smooth. Having a provable guarantee is a welcome contribution and furthermore the proof is derived from first principles by bounding the Hessian matrix elements which makes it very sound and natural.

(2) The derivation of a tighter bound $L \leq 4\|M\|_2 \sum_{k=1}^P\left\|G_k\right\|_2^2$ and the detailed comparison in Appendix C demonstrates that this bound is provably never looser, and often strictly tighter, than prior results from Gu et al. (2021) and Liu et al. (2025). The bound's explicit dependence on the sum of squared generator norms, rather than the maximum norm or observable decomposition, captures the individual contributions of each gate.

(3) In my view, this is a cool strength of the paper is the potential link between the saturation of landscape curvature and the saturation of ansatz expressibility (referring to Figure 1c). The proposal that the stabilization of the ratio \tilde{L}_{\rm max}/L_{ \rm upper} serves as a geometric signature of inefficient overparameterization is not something I have read before and maybe it can serve as tool for ansatz design. Now, the paper establishes a geometric view of barren plateaus as follows: it shows empirically that as the number of qubits n increases, the true maximum curvature ($\tilde{L}_{max}$) vanishes exponentially, much faster than the theoretical bound $L_{upper}$.This confirms that the entire landscape, including its "curviest" regions, is flattening out—a second-order signature of the barren plateas.
The interest finding which is illustrated in Figure 1c, comes from fixing the number of qubits and increasing the circuit depth P and the authors observe that the ratio of the true curvature to their theoretical bound ($\tilde{L}_{max}/L_{upper}$) eventually stabilizes. This observation, which should be further tested, is key since it is known that when a circuit is underparameterized, each new parameter adds significant representative power increasing its relative curvature. However, once the circuit's expressibility saturates for a given number of qubits, adding more layers and parameters yields diminishing returns. These new parameters become redundant, and their primary effect is to deepen the circuit, pushing it further into the barren plateau regime without improving its problem solving capacity. The paper seems to propose that the plateau in the curvature ratio ( $\tilde{L}_{\text {max }} / L_{\text {upper }}$ ) is the direct geometric manifestation of this inefficient overparameterization which may serve as a tool to inform the underlying hyper parameter choice in the construction of the ansatz in the first place. So, while knowing $L$ does not solve the problem of vanishing gradients, monitoring the relationship between the true curvature and the theoretical bound provides a concrete landscape-based signal which may allow a practitioner to identify the point at which adding more depth to their ansatz stops being productive and starts becoming a liability, increasing the risk of creating an untrainable, flat landscape. It would be interesting thus to see, how $L$ may inform such a circuit construction.

**Weaknesses:**

(1) The entire analysis is done in an idealized noiseless setting. The authors do acknowledges this by establishing the result as a theoretical baseline. However, this is a significant limitation. The primary challenge in practical VQA optimization stems from the stochastic nature of the objective function landscape induced by shot noise and hardware errors of all shorts. An analysis of L-smoothness in a setting where these dominant, non-smoothness-inducing effects are absent provides limited guidance for optimization on actual NISQ devices. The conclusions about stable, predictable curvature scaling may not hold when the optimizer interacts with a stochastic estimator of the objective function.

(2) The bound is potentially loose since the proof of Theorem 2 relies on the inequality $\|H\|_2 \leq\|B\|_2$, where $B_{k l}=4\|M\|_2\left\|G_k\right\|_2\left\|G_l\right\|_2$ is an element-wise upper bound on the Hessian matrix $H$. This step can introduce a substantial gap. The paper's own empirical results as shwon in Figure 1a show that the measured maximum curvature, $\tilde{L}_{\text {max }}$, is often only a small fraction of the theoretical upper bound $L_{\text {upper }}$. So while the bound correctly captures scaling, its significant looseness warrants a more detailed theoretical investigation maybe. The analysis could be strengthened by discussing the conditions under which the inequalities in the proof become equalities and what circuit physical properties (entanglement structure, parameter correlations) might govern the magnitude of this gap.

(3) The empirical ground truth for maximum curvature, $\tilde{L}_{\text {max }}$, is estimated by taking the maximum Hessian norm over 1000 random parameter samples. While Appendix D. 2 provides a reasonable justification for the stability of this estimate, this methodology cannot guarantee that the true global maximum of $\left\|\nabla^2 f(\theta)\right\|_2$ has been found in general. For that problems where the global optimizer is known are useful testbeds since hiigh-dimensional landscapes may contain rare and isolated regions of extreme curvature that are unlikely to be captured by uniform random sampling.

(4) The proposed heuristic is designed to set a single global learning rate. However, modern optimization heavily relies on adaptive methods like adam. So, while the existence of such a constant is proven, this framework is somewhat misaligned with the reality of modern, large-scale optimization unless we want to restrict ourselves to only talk about quantum optimization in isolation. As noted in the literature, e.g. https://arxiv.org/abs/2210.02418 for many typical problems, objective functions rarely satisfy uniform smoothness assumptions in a way that is practically useful their gradients may only be locally Lipschitz continuous, or the local curvature can vary dramatically across the parameter space. Of course, the VQA objective is usually globally L-bounded, as shown in this paper. But a global constant $L$, determined by the region of maximum curvature is excessively conservative for the majority of the landscape as far as using it for thelearning rate. Standard gradient descent with a step size derived from this global $L$ (e.g., $\eta \approx 1/L$) would take impractically small steps thus leading to slow convergence. This is precisely why SOTA optimizers really care to account for local geometry. The paper's proposed learning rate heuristic, while nice in principle, still provides a global rate, which does not align with modern optimization paradigms. The analysis would be significantly strengthened by contextualizing its findings within more modern frameworks, such as local or relative smoothness of the VQA objective in this sense.

**Questions:**

(1) How do you expect the main results and particularly the predictable linear scaling of curvature with depth, to change in the presence of realistic shot noise and hardware noise? This is super crucial. Does the concept of L-smoothness remain a useful descriptor for the stochastic objective function that an optimizer actually interacts with?

(2) Could you provide more theoretical insight into the large gap between the derived upper bound $L_{\text {upper }}$ and the empirically observed $\tilde{L}_{\text {max }}$ ? Does this gap depend on properties not captured by the bound, such as the circuit's entanglement capacity or the locality of the observable?

(3) The trigonometric polynomial proof route in Appendix A. 6 bounds the Fourier coefficients as $\left|d_\omega\right| \leq\|M\|_2$. Given that these coefficients have a specific structure ( $d_\omega=\left\langle u_\omega\right| M\left|v_\omega\right\rangle$ ), could a more refined analysis that does not resort to this uniform worst-case bound yield a tighter overall smoothness constant? These trigonometric polynomials, note, are actually Hermitian trigonometric polynomials in $d$ complex variables and the optimization takes place over the torus $\mathbb{T}^d$. does this not induce some "structure" to be exploited so as to further bound $L$?

(4) Regarding the learning rate heuristic, would it be more effective to use the calibrated effective smoothness constant, $L_{\rm  effective}$, to rescale the global learning rate of an adaptive optimizer like adam, rather than using it directly in a vanilla SGD context?

**Details Of Ethics Concerns:**

No ethics concerns.

---

> ### Author Response · Authors · 2025-11-20
>
> We thoroughly appreciate the detailed and in-depth review.
>
> **Noise (W1 & Q1)**: We have conducted a set of noisy VQE experiments. Please see Global **Point 3** and **revised Appendix F**.
>
> **Bound Looseness (W2 & Q2)**: Please see Global **Point 6** where we clarify that this looseness is a key finding (second-order barren plateau) and our heuristic is designed to solve it.
>
> **Sampling (W3)**: Please refer to Global **Point 5**.
>
> **Adaptive Optimizers (W4 & Q4)**: We clarify that our learning rate heuristic is **fully compatible with adaptive methods like Adam**. Please see Global **Point 4**.
>
> Finally for **Q3** this is a very intriguing question. A more refined, circuit-dependent analysis that exploits the MTP structure is a fascinating and non-trivial direction for future work. Our current work establishes the general bound and then focuses on an empirical correction (the $\kappa(n)$ heuristic) for this gap, but we agree a tighter analytical bound is a valuable goal (**Point 8**).
>
> *Is there anything else we can provide or clarify that would help you feel more confident about our paper?*

---

> > ### Author Response · Authors · 2025-12-01
> >
> > **The revised manuscript is now uploaded**. We have directly addressed your concerns regarding the "idealized noiseless setting" and the use of "adaptive methods" by adding Section 5.4 and Appendix F (Fig 11, Table 3). We demonstrate that our heuristic remains robust in noisy environments (depolarizing noise + shots) and enables Adam to achieve convergence rates competitive with the computationally expensive Quantum Natural Gradient (QNG) optimizer. We have also clarified that our heuristic is compatible with adaptive methods and expanded the discussion on $\kappa(n)$ in section 6.
> >
> > We appreciate the thorough review and believe we have addressed all core concerns.

---

### Official Review · Reviewer_oVNx · 2025-10-31

**Soundness:** 3
**Presentation:** 3
**Contribution:** 2
**Rating:** 4
**Confidence:** 4

**Summary:**

This work explores the stability of training Variational Quantum Algorithms (VQA). In the theoretical section, an explicit upper-bound of the L-smoothness parameter is derived. The numerical experiments investigate the tightness of their bound with respect to circuit depth and number of qubits and the connection of L-smoothness to expressability. In the last part, they propose a method to choose the learning rate such that stable training of the circuit is ensured, which is again numerically evaluated. The authors find that their bound is tight, up to a constant, for large enough circuit depth.

**Strengths:**

The paper is well-written and the results are presented in a clear fashion. Furthermore, there is a good connection between the numerical experiments and they represent a nice application of the theoretical results.

**Weaknesses:**

My main concern are the novelty and scope of the contribution. The improvement compared to [Liu et al., 2025](https://arxiv.org/pdf/2210.06723) seems incremental and the derivation does not appear to require elaborate tools. According to my understanding, Lemma 1 is a well-known fact, see e.g.
[Schuld et al., 2021](https://journals.aps.org/pra/abstract/10.1103/PhysRevA.103.032430).

The theoretical section would benefit from an investigation of the theoretical dependence of the smoothness on the number of qubits. Since results like [Holmes et al., 2022](https://arxiv.org/pdf/2101.02138) suggest that the decay of the L-smoothness may be exponential in $n$, the regime in which the smoothness scales proportional to the bound may only be reached after an exponential number of gates, potentially resulting in poor choices of learning rate.

Finally, overparametrization is not properly addressed. I agree with the observation that the VQE reaches an amount of parameters in the numerical experiment sufficient for exploring the Hilbert space. However, the work of [Larocca et al., 2023](https://arxiv.org/pdf/2109.11676) finds that the loss landscape resulting from sufficient overparametrization mitigates spurious local minima. This could be investigated by exploring lower-bounds of the Hessian.

**Questions:**

- How does the performance of GD with fine-tuned learning rate perform compared to the algorithm in [Liu et al., 2025](https://arxiv.org/pdf/2210.06723)?
- Is it possible give more theoretical insights about the dependence on the number of qubits? The average trace of the Hessian should correspond to the sum of the variance terms derived in [Holmes et al., 2022](https://arxiv.org/pdf/2101.02138).
- Is it possible to give lower-bounds for the Hessian? This could have implications for global convergence of the objective, using the Polyak-Lojasiewicz condition.

---

> ### Author Response · Authors · 2025-11-20
>
> We deeply appreciate the thorough and constructive feedback.
>
> Regarding **W1** we appreciate the opportunity to clarify our work's core theoretical contributions. We kindly refer you to **Point 1 and Point 2** in the global message.
>
> **Qubit Scaling (W2 & Q2)**: Please see Global **Point 6** where we clarify the distinction between our work and papers focused on first-order information.
>
> **Overparameterization**: Please see Global **Point 7** where we clarify our work’s contribution to connecting overparameterization with curvature.
>
> **Comparison to Liu et al. (Q1)**: Please see Global **Point 4**.
>
> **Lower Bounds (Q3)**: We consider this natural and interesting for future work (see Global **Point 8**).
>
> *Is there anything else we can provide or clarify that would help you feel more confident about our paper?*

---

> > ### Comment · Reviewer_oVNx · 2025-11-23
> > **response**
> >
> > Thanks for your detailed clarifications, I have revised my score accordingly.

---

> > > ### Author Response · Authors · 2025-12-01
> > >
> > > **The revised manuscript is now uploaded**. We have directly addressed your comments regarding overparameterization and the theoretical dependence on qubit count. Specifically, in Section 5.2, we explicitly link our observed curvature saturation to recent work on Dynamical Lie Algebras (Larocca et al., 2023) as a geometric signature of inefficient overparameterization. Furthermore, in Section 6 we expand discussion of the scaling factor $\kappa(n)$ as a second-order signature of the barren plateau phenomenon. We thank the reviewer for the thorough feedback which has strengthened our work.

---

### Official Review · Reviewer_uhHq · 2025-10-31

**Soundness:** 2
**Presentation:** 2
**Contribution:** 2
**Rating:** 4
**Confidence:** 4

**Summary:**

This paper tackles a foundational optimization question for VQAs: do their objective landscapes admit a global Lipschitz-smooth (L-smooth) constant that is tight enough to inform practice? The authors first prove that any VQA objective is a multivariate trigonometric polynomial (MTP) via an induction over gates (Lemma 1), and then derive a closed-form Hessian bound. Empirically, the paper shows the maximum curvature scales linearly with depth and exhibits a plateau that the authors use to build a calibrated learning-rate heuristic; small-scale VQE studies (1-4 qubits) indicate faster and more stable convergence under this calibration.

**Strengths:**

1. Table 1 clearly contrasts prior L-smoothness bounds (assumptions, formulas, and tightness), and argues the new bound is never looser, with up to a factor P improvement over other comparison methods. This helps readers situate contributions precisely.
2. Lemma 1 formalizes VQA objectives as finite Fourier series, motivating global smoothness and enabling multiple proof routes (generator-norm vs. Fourier-frequency). This is simple, general, and pedagogically valuable.
3. Derivations and settings are documented; code is provided, with software versions and experiment details.

**Weaknesses:**

1. The manuscript argues that tighter L helps gradient-based optimization but does not validate on real-world quantum datasets or hardware, and the experiments use small qubit counts; this limits the empirical case for broader utility. Consider adding larger-n simulations or a hardware study to demonstrate the robustness of the calibration heuristic and the curvature-depth scaling.
2. The plateau and scaling claims rely on random parameter sampling (S≈1000) to estimate $L_{max}$. Although convergence of estimates is reported, an optimization-based or certified bound on estimation error would increase confidence.
3. The learning-rate heuristic is tested with SGD/Adam on VQE only. It would be informative to compare against natural gradient / QNG or curvature-aware schedules to show the heuristic’s added value beyond simple step-size tuning.
4. Minor clarification issue: It would be better to double-check the equation and notation in the background introduction.

**Questions:**

1. You sample 1,000 parameter points to approximate the maximum curvature. What confidence guarantees (e.g., PAC-style) can you provide for the reported plateaus? Could you add a global-optimization routine (even on toy instances) to benchmark the sampler’s recall?
2. Your Lemma 1 proof covers single-parameter gates. How do your conclusions extend to multi-parameter exponentials or parameter sharing across layers? Is the MTP structure and the bound unchanged after standard decompositions?

---

> ### Author Response · Authors · 2025-11-20
>
> We thank the reviewer for the thorough and helpful review.
>
> **Noise & Hardware (W1)**: We have added a new set of VQE experiments under NISQ-like environments. Please see Global **Point 3** and **Appendix F**.
>
> **Sampling (W2 & Q1)**: The justification for our sampling methodology and the resulting stability data are detailed in Global **Point 5**.
>
> **QNG Comparison (W3)**: We agree and **have added QNG comparisons** showing our method is competitive but cheaper. Please see Global **Point 4**.
>
> **Multi-parameter gates (Q2)**: Please refer to Global **Point 7** for clarifications.
>
> *Is there anything else we can provide or clarify that would help you feel more confident about our paper?*

---

> > ### Comment · Reviewer_uhHq · 2025-11-27
> >
> > Thank you for your response.

---

> > > ### Author Response · Authors · 2025-12-01
> > >
> > > **The revised manuscript is now uploaded**. We have added **Section 5.4 and Appendix F (Table 3, Fig. 11)** to directly address your concerns regarding noise and the QNG baseline. We also added a Remark on Generality in Corollary 1  to **clarify the application to multi-parameter gates**. We thank you for the detailed review and believe all concerns are now addressed.

---

### Official Review · Reviewer_Huo3 · 2025-11-01

**Soundness:** 3
**Presentation:** 3
**Contribution:** 3
**Rating:** 6
**Confidence:** 4

**Summary:**

The paper proves that objectives of parametrized quantum circuits are globally L-smooth and gives an explicit upper bound on a smoothness constant. They compare against prior bounds and claim their expression is never looser and often tighter. Experiments on simulated circuits estimate the maximum Hessian norm by random sampling. It finds that curvature scales linearly with depth and observable norm, and that the ratio plateaus with depth. Th authors also propose a calibrated learning-rate heuristic.

**Strengths:**

1. the paper give a clear and tighter bounds on a smoothness with detailed derivations.
2. The results are informative diagnostics for ansatz design.

**Weaknesses:**

1. Global smoothness and upper bounds have been shown before; the main contribution is a tighter constant and cleaner derivation.
2. The experiments show the ratio plateaus with depth, but there is lack of analytically analysis.
3. Empirics are classical simulations on small widths and depths, with curvature estimated by sampling 1000 random parameter vectors.

**Questions:**

1. What are matching lower bounds on the global maximum curvature for realistic VQAs?
2. What will be the cost of curvature estimation? Can we reduce it with some estimators?
3. Can we prove concentration of the Hessian spectral norm for broad random ansatz ensembles?

---

> ### Author Response · Authors · 2025-11-20
>
> We appreciate the questions and opportunity to clarify our work's main contribution.
>
> **Global smoothness/Novelty (W1)**: We have revised the Introduction to clarify that our contribution is not merely the tightest bound, but the application of this bound to derive geometric findings. We ask to see **Points 1 & 2** in the global message.
>
> **Analytical Analysis (W2)**: **We kindly refer to Point 6.**
>
> **Sampling Effectiveness (W3)**: Evidence for the effectiveness of our sampling methodology (W3) is written in **Point 5**.
>
> **Cost of curvature estimation (Q2)**: Please see **Point 5** for details on the cost of curvature estimation. Our heuristic is designed to mitigate this cost.
>
> **Lower Bounds (Q1 & Q3)**: Our work focuses on the Hessian's upper bound ($L$) for optimization stability. Analyzing the lower bound ($\mu$) are the natural next steps. We view our work as laying the foundation for this future 'L/$\mu$' and concentration analysis by providing the tightest known '$L$'. We consider this outside the scope of the current contribution. **Point 8**.
>
> *Is there anything else we can provide or clarify that would help you feel more confident about our paper?*

---

> > ### Comment · Reviewer_Huo3 · 2025-11-26
> >
> > I would like to thank the authors for their detailed answers. It solves my questions. Given its current form, I would like to maintain my score.

---

> > > ### Author Response · Authors · 2025-12-01
> > >
> > > The **revised manuscript is now uploaded**. To address your concerns we have clarified the contributions in the revised Introduction. Furthermore, we expanded the Discussion to address your question on lower bounds and Polyak-Lojasiewicz (PL) conditions. We thank you for the detailed review and believe all concerns are now addressed.

---

### Official Review · Reviewer_Xmst · 2025-11-01

**Soundness:** 3
**Presentation:** 3
**Contribution:** 3
**Rating:** 6
**Confidence:** 3

**Summary:**

This paper aims to formalize and strengthen the theoretical understanding of the optimization landscapes underlying variational quantum algorithms (VQAs). It establishes that the objective functions of VQAs are globally L-smooth and derives a new upper bound on the smoothness constant, $L \le 4|M|^2 \sum_k |G_k|_2^2$, which is shown to be tighter, or at least no looser, than previous results. The authors further relate this bound to the geometry of quantum circuits, demonstrating that the curvature scales predictably in accordance with this bound and saturates at the high-expressibility limit, which they interpret as a geometric indicator of inefficient overparameterization. Finally, they propose a heuristic for setting near-optimal learning rates based on this analysis and validate it across multiple VQE benchmarks.

**Strengths:**

- The derivation of the bound looks clean and well-written. The bound holds for general circuits, and the authors provide an extensive comparison with previous results.

- They connect the bound with the geometry of the loss landscape and draw practical heuristics for setting the learning rate in training VQCs. Their claims are supported by extensive numerical experiments.

**Weaknesses:**

Both the bound and the numerical experiments assume an ideal, noise-free regime. While deriving a theoretical bound under noise would be challenging, numerical experiments with a realistic noise model (or on real quantum hardware via gradient computed through parameter-shift rule) could be feasibly conducted. Such experiments would help ground the insights and proposed heuristics in more practical settings.

**Questions:**

How would the proposed (heuristic) optimal learning rate compare to conventional methods with dynamic learning rate scheduling, or even to parameter-free methods such as [1] or [2]?

[1] Orabona, Francesco, and Tatiana Tommasi. "Training deep networks without learning rates through coin betting." Advances in neural information processing systems 30 (2017).

[2] Defazio, Aaron, and Konstantin Mishchenko. "Learning-rate-free learning by d-adaptation." International Conference on Machine Learning. PMLR, 2023.

---

> ### Author Response · Authors · 2025-11-20
>
> We thank you for the valuable feedback.
>
> “Numerical experiments assume an ideal, noise-free regime”:
>
> We agree this is important and have extended our VQE experiments to include noise models in **Appendix F**. See the global response **Point 3**.
>
> Dynamic/Parameter-free methods: We have clarified in the revision that our heuristic is a base calibration compatible with dynamic schedulers. We prioritized comparing against QNG (now added in Appendix F) as it is the standard curvature-aware baseline in Quantum ML. We refer to **Point 4**.
>
> *Is there anything else we can provide or clarify that would help you feel more confident about our work?*

---

> ### Comment · Reviewer_Xmst · 2025-11-26
> **Regarding the Depolarizing Channel in Appendix F**
>
> Thank you for the detailed responses to my concerns and questions. I’m glad to see that you have incorporated noisy quantum circuit simulation. However, in the current manuscript, I don’t see any mention of the depolarizing channel in Appendix F. Am I missing something, or is the uploaded version not yet the revised manuscript?

---

> > ### Author Response · Authors · 2025-11-26
> >
> > Thank you for the follow-up. You are correct, the current version is the original submission. We are currently waiting for all reviewers to respond so that we can upload a single, consolidated revision that addresses everyone's feedback at once.
> > However, since we have the results from the noise models ready, we wanted to share them with you immediately below rather than making you wait for the full upload.
> >
> >
> > **1. Noise Model Details**
> >
> > In the upcoming Appendix F, we detail the noise model used for the validation. We simulated a depolarizing noise channel applied after every gate, with error probabilities set to 0.1% for single-qubit gates and 1.0% for two-qubit gates. We coupled this with shot-based measurements (1000 shots per expectation value evaluation).
> >
> >
> > **2. Results (Comparison with QNG)**
> >
> > As requested, we compared our heuristic against the Quantum Natural Gradient (QNG). The results (see table below) demonstrate that our calibrated learning rate ("Optimal $\eta$") consistently outperforms standard learning rate settings and remains competitive with QNG.
> >
> >
> > | Optimizer | LR Setting        | Depth | Energy Epoch 10 | Energy Epoch 50 | Mean STD |
> > |-----------|-------------------|-------|------------------|------------------|----------|
> > | **Adam**  | Optimal η         | 6 | **-2.337** | -2.669 | 0.085 |
> > |           | High η            | 6 | -1.970 | **-2.724** | 0.109 |
> > |           | Low η             | 6 | -1.313 | -2.493 | 0.142 |
> > |           | Standard (0.001)  | 6 | -0.012 | -0.453 | 0.507 |
> > |           | Optimal η         | 8 | **-2.124** | **-2.497** | 0.082 |
> > |           | High η            | 8 | -1.662 | -2.444 | 0.107 |
> > |           | Low η             | 8 | -1.361 | -2.165 | 0.135 |
> > |           | Standard (0.001)  | 8 | -0.312 | -0.762 | 0.417 |
> > |           | Optimal η         | 10 | **-1.790** | **-2.315** | 0.079 |
> > |           | High η            | 10 | -1.494 | -2.163 | 0.113 |
> > |           | Low η             | 10 | -1.347 | -1.999 | 0.117 |
> > |           | Standard (0.001)  | 10 | 0.032 | -0.665 | 0.246 |
> > | **SGD**   | Optimal η         | 6 | **-2.015** | **-2.482** | 0.087 |
> > |           | High η            | 6 | -1.257 | -2.050 | 0.279 |
> > |           | Low η             | 6 | -0.671 | -1.947 | 0.138 |
> > |           | Standard (0.01)   | 6 | -0.356 | -1.155 | 0.297 |
> > |           | Optimal η         | 8 | **-1.600** | **-2.207** | 0.208 |
> > |           | High η            | 8 | -1.217 | -2.081 | 0.209 |
> > |           | Low η             | 8 | -0.574 | -1.564 | 0.110 |
> > |           | Standard (0.01)   | 8 | -0.274 | -1.216 | 0.267 |
> > |           | Optimal η         | 10 | -1.406 | **-1.938** | 0.089 |
> > |           | High η            | 10 | **-1.507** | -1.850 | 0.122 |
> > |           | Low η             | 10 | -0.179 | -1.240 | 0.277 |
> > |           | Standard (0.01)   | 10 | -0.303 | -1.309 | 0.098 |
> > | **QNG**   | Standard (0.01)   | 6 | -1.298 | -2.378 | 0.259 |
> > |           | Standard (0.01)   | 8 | -1.216 | -2.169 | 0.262 |
> > |           | Standard (0.01)   | 10 | -1.235 | -1.924 | 0.108 |

---

> ### Author Response · Authors · 2025-12-01
>
> The revised manuscript is **now uploaded**. We have directly addressed your concerns regarding the "ideal, noise-free regime" by adding **Section 5.4 and Appendix F (Fig 11, Table 3)**. Specifically, we implemented a noisy environment using depolarizing channels (0.1% error for single-qubit, 1.0% for two-qubit gates) coupled with 1000 shots-based measurements. We also added a direct comparison showing our heuristic is competitive with the Quantum Natural Gradient (QNG).

---

### Author Response · Authors · 2025-11-12

We thank all reviewers for the time and care they have dedicated to evaluating our submission. We appreciate the thoughtful feedback and many helpful suggestions provided across the reviews. We are currently in the process of carefully addressing all comments and will provide detailed, point-by-point responses and corresponding revisions shortly.

Thank you again for your valuable input and for helping us improve the clarity and impact of this work.

---

### Author Response · Authors · 2025-11-20
**New Experiments, Points 1-2**

We sincerely thank all reviewers for their thoughtful feedback. To ensure we address all comments, we first present major improvements addressing two core concerns (Noise & Baselines), followed by 8 Points responding to specific theoretical and methodological questions.

**Note on Revision**: All key changes and new results in the revised manuscript are highlighted in red for ease of review.

The most frequent concerns were:

1) The analysis is limited to an idealized, noiseless setting.
2) The learning rate heuristic was not compared against certain baselines.

We are pleased to report that we have conducted a new set of VQE experiments that directly address both of these points. These results are now detailed in **Appendix F** of the revised paper.

**Summary**:

**Noise & Shot-Noise**: We re-ran the 4-qubit VQE experiment using a realistic simulation with a depolarizing channel after each single-, double-qubit gate with shot-based measurements.

**New Baseline (QNG)**: We added a direct comparison against the commonly used curvature aware Quantum Natural Gradient (QNG).

**Findings**: Our learning rate heuristic remains highly effective. The calibrated SGD and Adam optimizers converge significantly faster and more stably than their un-calibrated counterparts (e.g., Adam optimal-lr reached a 322% lower energy than the standard-lr during the training period). Furthermore, our heuristic makes Adam/SGD competitive with the far more expensive QNG optimizer which requires computation of the full  $P \times P$ Fubini-Study metric at every step. Finally using the Optimal lr for Adam/SGD made the standard deviation over the 5 runs the smallest.

**Clarification of the core contributions**:
Our work addresses a critical gap in VQA research by turning established theoretical bounds into practical tools that guide optimization: revealing (i) second-order barren plateaus, (ii) linking curvature saturation to expressibility, (iii) enabling the $\kappa(n)$ learning-rate heuristic. Unlike prior studies, which treat theory and experiment separately, we provide a unified framework that translates abstract insights into actionable strategies for faster, more stable convergence in realistic quantum systems.

**1. Overview & Core Contributions (@Xmst @uhHq @Huo3 @ygvg @oVNx)**
We appreciate the request to clarify the main contributions. We have revised the Introduction to explicitly list the
core contributions as:

**C1. A new, tighter global smoothness bound.**

The closed-form upper bound:
$$L \le 4\Vert M \Vert_{2} \sum_{k=1}^{P} \Vert G_{k} \Vert_{2}^{2}$$
which is provably never looser and often significantly tighter (up to a factor of $P$ or $\sqrt{r}$) than prior results (Table 1, App. C).

**C2. A geometric interpretation of VQA landscapes.**

Using this bound, we uncover three new geometric properties:

(i) second-order barren plateau signature (curvature collapse with qubits)

(ii) predictable curvature growth with depth

(iii) connection between curvature saturation and expressibility / overparameterization.

**C3. A practical learning-rate calibration heuristic.**

Leveraging curvature-depth scaling, we derive a one-time learning rate calibration rule that significantly accelerates optimization.

**C4. Empirical validation (including new noisy experiments).**

Our VQE experiments, including new results added, show faster and more stable convergence for SGD/Adam, and competitive performance with the more expensive QNG optimizer.

These contributions go well beyond offering a tighter bound: they provide a geometric and practical framework for understanding and improving VQA optimization.

**2. Novelty & Theoretical Contributions (@oVNx @uhHq @Huo3 @ygvg)**

“Lemma 1 is known / incremental.” (**@oVNx**)

We agree that the MTP structure is known (Schuld 2021, Wierichs 2022). Our goal is not to claim novelty for the fact, but for how we use it: The MTP structure enables our new global Hessian bound. Our derivation of the bound in App A.6 uses this property. The core novelty lies in Findings 1-4:

**Key theoretical findings (not in prior work such as Liu 2025)**:

**Finding 1**: Second-order barren plateaus: curvature itself collapses exponentially with qubits.

**Finding 2**: Predictable curvature scaling: depth increases curvature in a structured, linear fashion (Fig. 1a, 1c).

**Finding 3**: Curvature saturation ↔ expressibility saturation: a new geometric marker for overparameterization (Fig. 1c).

**Finding 4**: Practical heuristic: $\kappa (n)$ uses curvature flattening to set learning rates effectively.
These offer a substantially deeper and more practical view of VQA landscapes than prior work focused on first-order gradients.
We now clarify this in the revised Introduction and Lemma 1.

---

### Author Response · Authors · 2025-11-20
**Points 3-5**

**3. Noise, NISQ Validation & New Experiments (@Xmst @uhHq @ygvg)**

“Results assume an idealized, noiseless regime.” (@Xmst @uhHq $ygvg)

Action taken: We agree this is important. We have now added a full suite of noisy VQE experiments in Appendix F.

**New experiment summary**:

* Noise model: depolarizing noise channel after every 1- and 2-qubit gate.
* Measurements: 1000-shot sampling.
* Systems: 4-qubit VQE.
* Baselines: Adam, SGD (both calibrated and uncalibrated), and the Quantum Natural Gradient (QNG) optimizer.

**Findings**:
* Our LR heuristic remains robust under noise highlighting the estimated curvature remains accurate and effective in NISQ like environments.
* Calibrated Adam/SGD converge significantly faster and more stably than uncalibrated versions.
* Our one-time calibration yields performance competitive with QNG, which requires a per-step evaluation of the full $P \times P$ Fubini–Study metric (Stokes 2020), far more computationally expensive.

These results directly address the reviewers’ concerns and demonstrate the heuristic’s NISQ relevance.

**4. Baselines & Optimizer Comparisons (QNG, LR-Free, Schedulers) (@Xmst @uhHq @oVNx @ygvg)**

On parameter-free methods (**@Xmst**): COCOB and D-Adaptation are not yet part of the VQA optimization literature, unlike QNG and Adam. Incorporating them requires careful, domain-specific implementation and tuning; we view this as an exciting direction for future work and consider the comparison with QNG more natural given its current use. Our heuristic is compatible with dynamic schedules; it provides the optimal base learning rate ($\eta$) from which schedules can decay.

On comparison with Liu et al. (2025) (**@oVNx**): Liu et al. focus on noise-based dynamics. Our work focuses on curvature-informed calibration. We view these as complementary approaches opposed to competing.

On Comparison with QNG (**@uhHq**): We added a direct comparison to QNG in Appendix F. Our calibrated Adam performs comparably to QNG while being orders of magnitude cheaper (QNG requires metric tensor inversion each step). This strongly validates the practical value of $\kappa (n)$.

“Modern optimization relies on adaptive methods. Do you use the calibrated effective smoothness constant to rescale the global learning rate of an adaptive optimizer like adam (**@ygvg**)”
This is precisely what we did and what the heuristic is designed for. $\kappa (n)$ is not intended to replace adaptive methods but to be fully compatible with them. Its primary goal is to solve the challenge of setting their global learning rate hyperparameter, $\eta$. We kindly refer the reviewer to our experiments in Section 5.3, Figure 10, and Table 2. These results explicitly show our heuristic being used to calibrate the learning rate for the ADAM optimizer. The noisy experiments in Appendix F show similar trends.

**5. Curvature Estimation & Sampling Robustness (@uhHq @Huo3 @ygvg)**

“1000 samples may miss rare curvature peaks. Provide guarantees.” (**@uhHq @ygvg**)

We appreciate this question. Our goal is not to compute the global maximum curvature, provably intractable (Bittel & Kliesch 2021), but to obtain a stable and practical estimate.

**Justification (expanded in App. D.2)**:
* Practical sufficiency: Calibration based on this sample yields faster convergence compared to smaller/larger learning rates (Sec. 5.3, App. F) indicating the curvature estimate is accurate.
* Convergence: Fig. 5a shows $\tilde{L}_{max}$ stabilizing well before 1000 samples; extending to 4000 samples yields negligible change.
* Stability: Ten independent 1000-sample runs (Fig. 7b) yield mean 3.210 ± 0.050, demonstrating low variance.
* Feasibility: Global optimization or PAC-style bounds are infeasible due to the exponential state vector (@uhHq).


Thus, our sampling-based estimate is stable, converged, and empirically validated.

Cost reduction (**@Huo3**): The cost on QPU is described in App B.2.3. Our heuristic is designed to overcome the computational burden of large Hessian computation. Instead of computing $L_{max}$ for the deep $P_{target}$ circuit, we compute it once for a much shallower $P_{cal}$ circuit. We use this cheap calibration to find the stable $\kappa(n)$ ratio, which then allows us to predict the curvature for the deep circuit. As we show in Appendix F.1, this provides a quadratic computational saving while achieving a prediction error of only 6-9% (Fig 8).

---

### Author Response · Authors · 2025-11-20
**Points 6-8 and Closing Statement**

**6. Qubit Scaling, Barren Plateaus & the κ(n) Heuristic (@oVNx @Huo3 @ygvg)**

“Provide theoretical insight into smoothness vs qubit number.” (**@oVNx**)

This is a central theme of our paper. Our empirical analysis (Fig. 1a, 1b, 5a) shows the normalized curvature $L_{max}$  / $L_{upper}$ collapses with qubits. This is a second-order barren plateau, complementing prior first-order analyses (McClean 2018; Holmes 2022) focused on vanishing first-order gradients. This finding directly motivates the need for $\kappa (n)$: upper bounds would yield overly small learning rates. $\kappa (n)$ provides a correction factor that captures how curvature flattens with increasing qubits.

“Regime where bound is proportional to curvature may require exponential gates. Resulting in poor choices of learning rate” (**@oVNx**)

Our heuristic is designed to tackle this problem. Our empirical results make the same point: curvature saturates, and the bound can become loose. Our contribution lies in:
* identifying this saturation empirically,
* linking it to expressibility (Fig. 1c), and
* providing a heuristic to manage it.

A full theoretical derivation of $\kappa (n)$ dependence is excellent future work (App. F.3).

“The experiments show the ratio plateaus with depth, but there is lack of analytically analysis. (**@Huo3**)”

A full analytical derivation for the plateau effect (stabilization of $L_{max} / L_{upper}$) is a significant theoretical challenge. Our work provides the empirical and geometric characterization of this phenomenon. We show it occurs in lockstep with expressibility saturation (Fig 1c), providing a new, second-order geometric signature for it. As we discuss in our new Appendix F.3, a formal proof is a major undertaking for future work.

Bound Tightness (**@ygvg**): We clarify in Section 5 that the growing gap between the bound and true curvature is a key finding, it is the geometric signature of the barren plateau. Our heuristic is the tool designed to empirically measure and correct for this gap. This is precisely why our heuristic is necessary. A naive LR derived from $L_{upper}$ would be "excessively conservative" (as the reviewer notes in W4). Our heuristic, $L_{effective} = \kappa(n) \cdot L_{upper}$, is our practical tool designed to explicitly empirically measure and correct for this $n$-dependent gap.

**7. Multi-Parameter Gates, Parameter Sharing & Expressibility (**@uhHq @oVNx**)**

“How does your analysis extend to multi-parameter gates?” (**@uhHq**)

This is addressed via Corollary 1 (Sec. 4.2). Multi-qubit rotations like $R_{xx}(\theta)$ are generated by a single Hermitian $G_k$, and the bound
$L \le 4\Vert M \Vert_{2} \sum_{k=1}^{P} \Vert G_{k} \Vert_{2}^{2}$
Still applies directly.

“What about parameter sharing?” (**@uhHq**)

Parameter sharing is compatible with the MTP structure (App. E), but the Hessian structure changes due to tied parameters. We now clarify this assumption in Sec. 4 and identify parameter sharing as a natural direction for future work, especially for architectures like QCNNs (Cong 2019).

Overparameterization & expressibility (**@oVNx**)

We emphasize in **Section 5.2** that our work complements DLA-based theory (Larocca 2023): in Fig. 1c, expressibility saturation $D_{KL}$ coincides with curvature saturation. This complements overparameterization theory by offering an intuitive, curvature-based marker of when the ansatz becomes inefficient.

**8. Future Work, Limitations & Additional Clarifications (@all)**

We acknowledge and appreciate the reviewers’ suggestions. In the revised Discussion and Appendix F.3, we have added discussions on:

* Potential integration with parameter-free methods.
* The Polyak-Lojasiewicz (PL) condition and Hessian lower bounds.
* Theoretical investigation of the $\kappa(n)$ plateau.

**Closing Statement**
We deeply appreciate the reviewers’ insightful comments, which have helped us improve the work. Our added noisy experiments, new QNG baseline, and clarified theoretical framing directly address the main concerns. We believe these revisions strengthen both the practical relevance and conceptual contribution of our work, and we hope our clarifications resolve the concerns raised.
Thank you for your thoughtful evaluation.

---

### Author Response · Authors · 2025-12-01
**Revised Manuscript Uploaded: New Noisy Experiments, QNG Comparison, and Expanded Theoretical Discussions**

Dear Area Chair and Reviewers,


We have uploaded the revised manuscript. In response to the reviewers' feedback, we have made significant updates to validate our findings in realistic settings and clarify the theoretical scope of our work. Below is a summary of the key additions and revisions:

**1. Major Addition: Validation in Noisy Environments & QNG Comparison (Section 5.4 & Appendix F)**

To address concerns regarding idealized settings, we added a comprehensive evaluation in Section 5.4 and Appendix F (including Figure 11 and Table 3).

* **Setup:** We implemented a noisy environment using depolarizing error channels (0.1% single-qubit, 1.0% two-qubit) coupled with shot-based measurements (1000 shots).
* **New Baseline:** We compared our method against the Quantum Natural Gradient (QNG).
* **Result:** We demonstrate that our heuristic remains robust in NISQ environments. It enables Adam and SGD to achieve convergence rates competitive with the computationally expensive QNG optimizer, while offering significantly lower computational overhead.

**2. Clarification on Generality and Proof Structure**
* **Proof Intuition:** We refined the introduction of our proof to explicitly state that we leverage the established trigonometric polynomial structure of VQA objectives (Wierichs et al., 2022) to derive the curvature bound.
* **Generality:** We added a "Remark on Generality" in Corollary 1 to clarify that our bound extends naturally to multi-parameter gates (via the spectral norm of $G$) and architectures with parameter sharing, such as QCNNs (where the objective retains its trigonometric polynomial structure).

**3. Justification of Methodology**
* **Sampling Stability:** We expanded Section 5.1 and Appendix D.2 to justify our use of $S=1000$ samples. We clarify that while global maximization is intractable, our convergence analysis confirms this sample size provides a stable estimate of the characteristic maximum curvature with low variance across independent runs.
* **Adaptive Optimizers:** We explicitly clarified that our heuristic is fully compatible with adaptive methods, where the calculated $\eta_{global}^*$ serves as the optimal base learning rate for optimizers like Adam.

**4. Expanded Discussion on Interpretability and Future Work**
* **Overparameterization:** In Section 5.2, we added a discussion linking the observed saturation of curvature to recent theoretical work on Dynamical Lie Algebras (Larocca et al., 2023), framing our result as a complementary, landscape-based signature of inefficient overparameterization.
* **Future Directions:** We expanded the Discussion (Section 6) to outline pathways for future theoretical work, including using random matrix theory to derive the plateau factor $\kappa(n)$ and investigating lower bounds to establish Polyak-Lojasiewicz (PL) conditions for global convergence.

We believe these revisions directly address the reviewers' feedback and significantly strengthen the manuscript.

Best regards,
The Authors

---

### Meta-Review · Area_Chair_fbJP · 2025-12-24

**Summary:**

This submission studies the global Lipschitz smoothness of variational quantum algorithm (VQA) objective landscapes. Leveraging the trigonometric polynomial structure of parameterized quantum circuits, the authors derive an upper bound on the Hessian spectral norm, showing it scales with the sum of squared generator norms. The submission also proposes a one-time learning-rate calibration heuristic for gradient-based optimizers. Numerical simulations exhibit the practical usefulness of the heuristic.

Overall, the five reviewers provided largely borderline scores, with concerns mainly centered on the strength and novelty of the theoretical results, the tightness and correctness of the derived bounds, and whether the proposed learning-rate heuristic offers sufficiently new insight beyond existing smoothness-based optimization analyses.

In light of these borderline assessments, I conducted a careful reading of the manuscript and identified two critical issues that were not raised by the reviewers, as detailed below. These issues affect the validity of the derived bound. Given that they are not minor technical oversights that could be resolved through small revisions, I believe the submission cannot be accepted in its current form, and I therefore recommend rejection.

**Reviewer Concerns:**

The rebuttal clarified the intended scope and positioning of the theoretical results, emphasizing that the contribution concerns a global Lipschitz smoothness bound rather than a local or purely empirical analysis. The authors also provided additional intuition for the Hessian bound, clarified the motivation and intended use of the one-time learning-rate calibration heuristic, and acknowledged presentation issues such as notation consistency and readability, with a commitment to revise these aspects.

Several concerns still remain after the rebuttal. First, while the rebuttal reiterates that the bound is global, it does not convincingly address whether the resulting bound is overly loose in realistic settings. Next, it remains unclear to what extent the learning-rate calibration heuristic is genuinely grounded in the theoretical analysis, as opposed to being an empirically motivated rule that happens to align with the bound. Although multiple numerical experiments are presented, concerns remain as to whether they are sufficient to support the paper’s broader claims across different architectures and problem classes.

Besides the remaining concerns, there are two other critical issues:
1. On page 17, the final inequality (Line 868) appears to omit a summation term over frequency components $\Omega$. If this understanding is correct, this term typically scales exponentially with the number of parameterized gates with $|\Omega|\sim O(\exp(P))$. As a result, the upper bound should be $|H_{kl}|\leq 4\exp(P)\|G_k\|_2\|G_l\|_2\|M\|_2$. Omitting it substantially alters the claimed bound and directly affects the validity of the main result.

2. The authors repeatedly claim that their results advance the bounds of Liu et al. However, when examining the bounds reported in Table 1 in isolation without considering the potential omission of the exponential term, this improvement does not hold in practical quantum computing settings. In particular, for standard parameterized circuits implemented with rotational Pauli gates, the proposed bound reduces to the same scaling as that of Liu et al.

**Reviewer Scores:**

Given the rebuttal and the subsequent discussion, it is unlikely that any reviewer would have increased their score. While some clarifications were provided, the discussion reveals deeper issues affecting the correctness and practical significance of the main theoretical results, including problems not identified in the original reviews. Had reviewers participated fully in the discussion, these issues would likely have reinforced their initial concerns, leading scores to remain unchanged or potentially decrease rather than improve.

---

### Decision · Program_Chairs · 2026-01-26

Reject